# STOCHASTIC DIFFERENTIALLY PRIVATE AND FAIR LEARNING

**Andrew Lowy**
University of Southern California
`lowya@usc.edu`

**Devansh Gupta**[*]
Indraprastha Institute of Information Technology, Delhi
`devansh19160@iiitd.ac.in`

**Meisam Razaviyayn**
University of Southern California
`razaviya@usc.edu`

## ABSTRACT

Machine learning models are increasingly used in high-stakes decision-making systems. In such applications, a major concern is that these models sometimes discriminate against certain demographic groups such as individuals with certain race, gender, or age. Another major concern in these applications is the violation of the privacy of users. While *fair learning* algorithms have been developed to mitigate discrimination issues, these algorithms can still *leak* sensitive information, such as individuals' health or financial records. Utilizing the notion of *differential privacy (DP)*, prior works aimed at developing learning algorithms that are both private and fair. However, existing algorithms for DP fair learning are either not guaranteed to converge or require full batch of data in each iteration of the algorithm to converge. In this paper, we provide the first *stochastic* differentially private algorithm for fair learning that is guaranteed to converge. Here, the term "stochastic" refers to the fact that our proposed algorithm converges even when *minibatches* of data are used at each iteration (i.e. *stochastic optimization*). Our framework is flexible enough to permit different fairness notions, including demographic parity and equalized odds. In addition, our algorithm can be applied to non-binary classification tasks with multiple (non-binary) sensitive attributes. As a byproduct of our convergence analysis, we provide the first utility guarantee for a DP algorithm for solving nonconvex-strongly concave min-max problems. Our numerical experiments show that the proposed algorithm *consistently offers significant performance gains over the state-of-the-art baselines*, and can be applied to larger scale problems with non-binary target/sensitive attributes.

## 1 INTRODUCTION

In recent years, machine learning algorithms have been increasingly used to inform decisions with far-reaching consequences (e.g. whether to release someone from prison or grant them a loan), raising concerns about their compliance with laws, regulations, societal norms, and ethical values. Specifically, machine learning algorithms have been found to discriminate against certain "sensitive" demographic groups (e.g. racial minorities), prompting a profusion of *algorithmic fairness* research (Dwork et al., 2012; Sweeney, 2013; Datta et al., 2015; Feldman et al., 2015; Bolukbasi et al., 2016; Angwin et al., 2016; Calmon et al., 2017; Hardt et al., 2016a; Fish et al., 2016; Woodworth et al., 2017; Zafar et al., 2017; Bechavod & Ligett, 2017; Kearns et al., 2018; Prost et al., 2019; Baharlouei et al., 2020; Lowy et al., 2022a). Algorithmic fairness literature aims to develop fair machine learning algorithms that output non-discriminatory predictions.

Fair learning algorithms typically need access to the sensitive data in order to ensure that the trained model is non-discriminatory. However, consumer privacy laws (such as the E.U. General Data Protection Regulation) restrict the use of sensitive demographic data in algorithmic decision-making.

---

[*]Work done as a visiting scholar at the University of Southern California, Viterbi School of Engineering.

These two requirements–*fair algorithms* trained with *private data*–presents a quandary: how can we train a model to be fair to a certain demographic if we don't even know which of our training examples belong to that group?

The works of Veale & Binns (2017); Kilbertus et al. (2018) proposed a solution to this quandary using secure *multi-party computation (MPC)*, which allows the learner to train a fair model without directly accessing the sensitive attributes. Unfortunately, as Jagielski et al. (2019) observed, *MPC does not prevent the trained model from leaking sensitive data*. For example, with MPC, the output of the trained model could be used to infer the race of an individual in the training data set (Fredrikson et al., 2015; He et al., 2019; Song et al., 2020; Carlini et al., 2021). To prevent such leaks, Jagielski et al. (2019) argued for the use of *differential privacy* (Dwork et al., 2006) in fair learning. Differential privacy (DP) provides a strong guarantee that no company (or adversary) can learn much more about any individual than they could have learned had that individual's data never been used.

Since Jagielski et al. (2019), several follow-up works have proposed alternate approaches to DP fair learning (Xu et al., 2019; Ding et al., 2020; Mozannar et al., 2020; Tran et al., 2021b;a; 2022). As shown in Fig. 1, each of these approaches suffers from at least two critical shortcomings. In particular, *none of these methods have convergence guarantees when mini-batches of data are used in training*. In training large-scale models, memory and efficiency constraints require the use of small minibatches in each iteration of training (i.e. stochastic optimization). Thus, existing DP fair learning methods cannot be used in such settings since they require computations on the full training data set in every iteration. See Appendix A for a more comprehensive discussion of related work.

**Our Contributions:** In this work, we propose a novel algorithmic framework for DP fair learning. Our approach builds on the non-private fair learning method of Lowy et al. (2022a). We consider a regularized empirical risk minimization (ERM) problem where the regularizer penalizes fairness violations, as measured by the *Exponential Rényi Mutual Information*. Using a result from Lowy et al. (2022a), we reformulate this fair ERM problem as a min-max optimization problem. Then, we use an efficient differentially private variation of stochastic gradient descent-ascent (DP-SGDA) to solve this fair ERM min-max objective. The main features of our algorithm are:

1. *Guaranteed convergence* for any privacy and fairness level, *even when mini-batches of data are used* in each iteration of training (i.e. stochastic optimization setting). As discussed, stochastic optimization is essential in large-scale machine learning scenarios. Our algorithm is the first stochastic DP fair learning method with provable convergence.

2. Flexibility to handle *non-binary* classification with *multiple (non-binary) sensitive attributes* (e.g. race and gender) under *different fairness notions* such as demographic parity or equalized odds. In each of these cases, our algorithm is guaranteed to converge.

*Empirically,* we show that our method *outperforms the previous state-of-the-art methods* in terms of fairness vs. accuracy trade-off across all privacy levels. Moreover, our algorithm is capable of training with mini-batch updates and can handle *non-binary target and non-binary sensitive attributes*. By contrast, existing DP fairness algorithms could not converge in our stochastic/non-binary experiment.

A byproduct of our algorithmic developments and analyses is *the first DP convergent algorithm for nonconvex min-max optimization*: namely, we provide an upper bound on the stationarity gap of DP-SGDA for solving problems of the form $\min_\theta \max_W F(\theta, W)$, where $F(\cdot, W)$ is non-convex. We expect this result to be of independent interest to the DP optimization community. Prior works that provide convergence results for DP min-max problems have assumed that $F(\cdot, W)$ is either (strongly) convex (Boob & Guzmán, 2021; Zhang et al., 2022) or satisfies a generalization of strong convexity known as the *Polyak-Łojasiewicz* (PL) condition (Yang et al., 2022).

## 2 PROBLEM SETTING AND PRELIMINARIES

Let $Z = \{z_i = (x_i, s_i, y_i)\}_{i=1}^n$ be a data set with non-sensitive features $x_i \in \mathcal{X}$, discrete sensitive attributes (e.g. race, gender) $s_i \in [k] \triangleq \{1, \ldots, k\}$, and labels $y_i \in [l]$. Let $\widehat{y}_\theta(x)$ denote the model predictions parameterized by $\theta$, and $\ell(\theta, x, y) = \ell(\widehat{y}_\theta(x), y)$ be a loss function (e.g. cross-entropy loss). Our goal is to (approximately) solve the empirical risk minimization (ERM) problem

$$\min_\theta \left\{ \widehat{\mathcal{L}}(\theta) := \frac{1}{n} \sum_{i=1}^n \ell(\theta, x_i, y_i) \right\} \tag{1}$$

in a fair manner, while maintaining the differential privacy of the sensitive data $\{s_i\}_{i=1}^n$. We consider two different notions of fairness in this work:[1]

**Definition 2.1** (Fairness Notions). Let $\mathcal{A} : \mathcal{Z} \to \mathcal{Y}$ be a classifier.

- $\mathcal{A}$ satisfies *demographic parity* (Dwork et al., 2012) if the predictions $\mathcal{A}(Z)$ are statistically independent of the sensitive attributes.
- $\mathcal{A}$ satisfies *equalized odds* (Hardt et al., 2016a) if the predictions $\mathcal{A}(Z)$ are conditionally independent of the sensitive attributes given $Y = y$ for all $y$.

Depending on the specific problem at hand, one fairness notion may be more desirable than the other (Dwork et al., 2012; Hardt et al., 2016a).

In practical applications, achieving exact fairness, i.e. (conditional) independence of $\widehat{Y}$ and $S$, is unrealistic. In fact, achieving exact fairness can be *impossible* for a differentially private algorithm that achieves non-trivial accuracy (Cummings et al., 2019). Thus, we instead aim to design an algorithm that achieves small *fairness violation* on the given data set $Z$. Fairness violation can be measured in different ways: see e.g. Lowy et al. (2022a) for a thorough survey. For example, if demographic parity is the desired fairness notion, then one can measure (empirical) demographic parity violation by

$$\max_{\widehat{y} \in \mathcal{Y}} \max_{s \in \mathcal{S}} \left| \hat{p}_{\widehat{Y}|S}(\widehat{y}|s) - \hat{p}_{\widehat{Y}}(\widehat{y}) \right|, \qquad (2)$$

where $\hat{p}$ denotes an empirical probability calculated directly from $(Z, \{\widehat{y}_i\}_{i=1}^n)$.

Next, we define differential privacy (DP). Following the DP fair learning literature in (Jagielski et al., 2019; Tran et al., 2021b; 2022)), we consider a relaxation of DP, in which only the *sensitive attributes* require privacy. Say $Z$ and $Z'$ are *adjacent with respect to sensitive data* if $Z = \{(x_i, y_i, s_i)\}_{i=1}^n$, $Z' = \{(x_i, y_i, s_i')\}_{i=1}^n$, and there is a unique $i \in [n]$ such that $s_i \neq s_i'$.

| Reference | Non-binary target? | Multiple fairness notions? | Convergence guarantee (poly. time)? | Guarantees with mini-batches? |
|---|:---:|:---:|:---:|:---:|
| *This work* | ✅ | ✅ | ✅ | ✅ |
| Jagielski et al. (2019) (post-proc.)* | ❌ | ❌ | N/A | ❌ |
| Jagielski et al. (2019) (in-proc.) | ❌ | ❌ | ❌ | ❌ |
| Xu et al. (2019) | ❌ | ❌ | ❌ | ❌ |
| Ding et al. (2020) | ❌ | ✅ | ❌ | ❌ |
| Mozannar et al. (2020) | ❌ | ✅ | ✅ | ❌ |
| Tran et al. (2021a) | ✅ | ❌ | ❌ | ❌ |
| Tran et al. (2021b) | ✅ | ✅ | ❌ | ❌ |
| Tran et al. (2022) | ✅ | ✅ | ❌ | ❌ |

Figure 1: Comparison with existing works. "Guarantee" refers to *provable* guarantee. N/A: the post-processing method of Jagielski et al. (2019) is not an iterative algorithm. *Method requires access to the sensitive data at test time. The in-processing method of Jagielski et al. (2019) is inefficient. The work of Mozannar et al. (2020) specializes to equalized odds, but most of their analysis seems to be extendable to other fairness notions.

**Definition 2.2** (Differential Privacy w.r.t. Sensitive Attributes). Let $\epsilon \geqslant 0$, $\delta \in [0, 1)$. A randomized algorithm $\mathcal{A}$ is $(\epsilon, \delta)$-*differentially private w.r.t. sensitive attributes $S$* (DP) if for all pairs of data sets $Z, Z'$ that are *adjacent w.r.t. sensitive attributes*, we have

$$\mathbb{P}(\mathcal{A}(Z) \in O) \leqslant e^{\epsilon} \mathbb{P}(\mathcal{A}(Z) \in O) + \delta, \qquad (3)$$

for all measurable $O \subseteq \mathcal{Y}$.

As discussed in Section 1, Theorem 2.2 is useful if a company wants to train a fair model, but is unable to use the sensitive attributes (which are needed to train a fair model) due to privacy concerns and laws (e.g., the E.U. GDPR). Theorem 2.2 enables the company to privately use the sensitive attributes to train a fair model, while satisfying legal and ethical constraints. That being said, Theorem 2.2 still may not prevent leakage of *non-sensitive* data. Thus, if the company is concerned with privacy of user data beyond the sensitive demographic attributes, then it should impose DP for all the features. Our algorithm and analysis readily extends to DP for all features: see Section 3.

Throughout the paper, we shall restrict attention to data sets that contain at least $\rho$-fraction of every sensitive attribute for some $\rho \in (0, 1)$: i.e. $\frac{1}{|Z|} \sum_{i=1}^{|Z|} \mathbb{1}_{\{s_i = r\}} \geqslant \rho$ for all $r \in [k]$. This is a reasonable

---

[1]Our method can also handle any other fairness notion that can be defined in terms of statistical (conditional) independence, such as equal opportunity. However, our method cannot handle all fairness notions: for example, false discovery rate and calibration error are not covered by our framework.

assumption in practice: for example, if sex is the sensitive attribute and a data set contains all men, then training a model that is fair with respect to sex and has a non-trivial performance (better than random) seems almost impossible. Understanding what performance is (im-)possible for DP fair learning in the absence of sample diversity is an important direction for future work.

## 3    PRIVATE FAIR ERM VIA EXPONENTIAL RÉNYI MUTUAL INFORMATION

A standard in-processing strategy in the literature for enforcing fairness is to add a regularization term to the empirical objective that penalizes fairness violations (Zhang et al., 2018; Donini et al., 2018; Mary et al., 2019; Baharlouei et al., 2020; Cho et al., 2020b; Lowy et al., 2022a). We can then jointly optimize for fairness and accuracy by solving

$$\min_{\theta} \left\{ \widehat{\mathcal{L}}(\theta) + \lambda \mathcal{D}(\widehat{Y}, S, Y) \right\},$$

where $\mathcal{D}$ is some measure of statistical (conditional) dependence between the sensitive attributes and the predictions (given $Y$), and $\lambda \geqslant 0$ is a scalar balancing fairness and accuracy considerations. The choice of $\mathcal{D}$ is crucial and can lead to different fairness-accuracy profiles. Inspired by the strong empirical performance and amenability to stochastic optimization of Lowy et al. (2022a), we choose $\mathcal{D}$ to be the Exponential Rényi Mutual Information (ERMI):

**Definition 3.1** (ERMI – Exponential Rényi Mutual Information). We define the exponential Rényi mutual information between random variables $\widehat{Y}$ and $S$ with empirical joint distribution $\hat{p}_{\widehat{Y}, S}$ and marginals $\hat{p}_{\widehat{Y}}$, $\hat{p}_S$ by:

$$\widehat{D}_R(\widehat{Y}, S) := \mathbb{E} \left\{ \frac{\hat{p}_{\widehat{Y}, S}(\widehat{Y}, S)}{\hat{p}_{\widehat{Y}}(\widehat{Y}) \hat{p}_S(S)} \right\} - 1 = \sum_{j \in [l]} \sum_{r \in [k]} \frac{\hat{p}_{\widehat{Y}, S}(j, r)^2}{\hat{p}_{\widehat{Y}}(j) \hat{p}_S(r)} - 1 \qquad \text{(ERMI)}$$

Theorem 3.1 is what we would use if *demographic parity* were the desired fairness notion. If instead one wanted to encourage equalized odds, then Theorem 3.1 can be readily adapted to these fairness notions by substituting appropriate conditional probabilities for $\hat{p}_{\widehat{Y}, S}$, $\hat{p}_{\widehat{Y}}$, and $\hat{p}_S$ in (ERMI): see Appendix B for details.[2] It can be shown that ERMI $\geqslant 0$, and is zero if and only if demographic parity (or equalized odds, for the conditional version of ERMI) is satisfied (Lowy et al., 2022a). Further, ERMI provides an upper bound on other commonly used measures of fairness violation: e.g.) (2), Shannon mutual information (Cho et al., 2020a), Rényi correlation (Baharlouei et al., 2020), $L_q$ fairness violation (Kearns et al., 2018; Hardt et al., 2016a) (Lowy et al., 2022a). This implies that any algorithm that makes ERMI small will also have small fairness violation with respect to these other notions. Lastly, (Lowy et al., 2022a, Proposition 2) shows that empirical ERMI (Theorem 3.1) is an asymptotically unbiased estimator of "population ERMI"–which can be defined as in Theorem 3.1, except that empirical distributions are replaced by their population counterparts.

Our approach to enforcing fairness is to augment (1) with an ERMI regularizer and privately solve:

$$\min_{\theta} \left\{ \text{FERMI}(\theta) := \widehat{\mathcal{L}}(\theta) + \lambda \widehat{D}_R(\widehat{Y}_\theta(X), S) \right\}. \qquad \text{(FERMI obj.)}$$

Since empirical ERMI is an asymptotically unbiased estimator of population ERMI, a solution to (FERMI obj.) is likely to generalize to the corresponding fair population risk minimization problem (Lowy et al., 2022a). There are numerous ways to privately solve (FERMI obj.). For example, one could use the exponential mechanism (McSherry & Talwar, 2007), or run noisy gradient descent (GD) (Bassily et al., 2014). The problem with these approaches is that they are inefficient or require computing $n$ gradients at every iteration, which is prohibitive for large-scale problems, as discussed earlier. Notice that we could *not* run noisy *stochastic* GD (SGD) on (FERMI obj.) because we do not (yet) have a statistically unbiased estimate of $\nabla_\theta \widehat{D}_R(\widehat{Y}_\theta(X), S)$.

Our next goal is to derive a *stochastic*, differentially private fair learning algorithm. For feature input $x$, let the predicted class labels be given by $\widehat{y}(x, \theta) = j \in [l]$ with probability $\mathcal{F}_j(x, \theta)$, where $\mathcal{F}(x, \theta)$ is differentiable in $\theta$, has range $[0, 1]^l$, and $\sum_{j=1}^l \mathcal{F}_j(x, \theta) = 1$. For instance,

---

[2] To simplify the presentation, we will assume that demographic parity is the fairness notion of interest in the remainder of this section. However, we consider both fairness notions in our numerical experiments.

$\mathcal{F}(x,\theta) = (\mathcal{F}_1(x,\theta), \dots, \mathcal{F}_l(x,\theta))$ could represent the output of a neural net after softmax layer or the probability label assigned by a logistic regression model. Then we have the following min-max re-formulation of (FERMI obj.):

**Theorem 3.2** (Lowy et al. (2022a)). *There are differentiable functions $\widehat{\psi}_i$ such that (FERMI obj.) is equivalent to*

$$\min_{\theta} \max_{W \in \mathbb{R}^{k \times l}} \left\{ \widehat{F}(\theta, W) := \widehat{\mathcal{L}}(\theta) + \lambda \frac{1}{n} \sum_{i=1}^{n} \widehat{\psi}_i(\theta, W) \right\}. \tag{4}$$

*Further, $\widehat{\psi}_i(\theta, \cdot)$ is strongly concave for all $\theta$.*

The functions $\widehat{\psi}_i$ are given explicitly in Appendix C. Theorem 3.2 is useful because it permits us to use *stochastic* optimization to solve (FERMI obj.): for any batch size $m \in [n]$, the gradients (with respect to $\theta$ and $W$) of $\frac{1}{m} \sum_{i \in \mathcal{B}} \ell(x_i, y_i; \theta) + \lambda \widehat{\psi}_i(\theta, W)$ are statistically unbiased estimators of the gradients of $\widehat{F}(\theta, W)$, if $\mathcal{B}$ is drawn uniformly from $Z$. However, when differential privacy of the sensitive attributes is also desired, the formulation (4) presents some challenges, due to the non-convexity of $\widehat{F}(\cdot, W)$. Indeed, *there is no known DP algorithm for solving non-convex min-max problems that is proven to converge*. Next, we provide the first such convergence guarantee.

### 3.1 NOISY DP-FERMI FOR STOCHASTIC PRIVATE FAIR ERM

Our proposed stochastic DP algorithm for solving (FERMI obj.), is given in Algorithm 1. It is a noisy DP variation of two-timescale stochastic gradient descent ascent (SGDA) Lin et al. (2020).

---

**Algorithm 1** DP-FERMI Algorithm for Private Fair ERM

---

1: **Input**: $\theta_0 \in \mathbb{R}^{d_\theta}$, $W_0 = 0 \in \mathbb{R}^{k \times l}$, step-sizes $(\eta_\theta, \eta_w)$, fairness parameter $\lambda \geqslant 0$, iteration number $T$, minibatch size $|B_t| = m \in [n]$, set $\mathcal{W} \subset \mathbb{R}^{k \times l}$, noise parameters $\sigma_w^2, \sigma_\theta^2$.
2: Compute $\widehat{P}_S^{-1/2}$.
3: **for** $t = 0, 1, \dots, T$ **do**
4:     Draw a mini-batch $B_t$ of data points $\{(x_i, s_i, y_i)\}_{i \in B_t}$
5:     Set $\theta_{t+1} \leftarrow \theta_t - \frac{\eta_\theta}{|B_t|} \sum_{i \in B_t} [\nabla_\theta \ell(x_i, y_i; \theta^t) + \lambda(\nabla_\theta \widehat{\psi}_i(\theta_t, W_t) + u_t)]$, where $u_t \sim \mathcal{N}(0, \sigma_\theta^2 \mathbf{I}_{d_\theta})$.
6:     Set $W_{t+1} \leftarrow \Pi_{\mathcal{W}} \left( W_t + \eta_w \left[ \frac{\lambda}{|B_t|} \sum_{i \in B_t} \nabla_w \widehat{\psi}_i(\theta_t, W_t) + V_t \right] \right)$, where $V_t$ is a $k \times l$ matrix with independent random Gaussian entries $(V_t)_{r,j} \sim \mathcal{N}(0, \sigma_w^2)$.
7: **end for**
8: Pick $\hat{t}$ uniformly at random from $\{1, \dots, T\}$.
9: **Return:** $\hat{\theta}_T := \theta_{\hat{t}}$.

---

Explicit formulae for $\nabla_\theta \widehat{\psi}_i(\theta_t, W_t)$ and $\nabla_w \widehat{\psi}_i(\theta_t, W_t)$ are given in Theorem D.1 (Appendix D). We provide the privacy guarantee of Algorithm 1 in Theorem 3.3:

**Theorem 3.3.** *Let $\epsilon \leqslant 2 \ln(1/\delta)$, $\delta \in (0, 1)$, and $T \geqslant \left( n \frac{\sqrt{\epsilon}}{2m} \right)^2$. Assume $\mathcal{F}(x, \cdot)$ is $L_\theta$-Lipschitz for all $x$, and $|(W_t)_{r,j}| \leqslant D$ for all $t \in [T], r \in [k], j \in [l]$. Then, for $\sigma_w^2 \geqslant \frac{16T \ln(1/\delta)}{\epsilon^2 n^2 \rho}$ and $\sigma_\theta^2 \geqslant \frac{16 L_\theta^2 D^2 \ln(1/\delta) T}{\epsilon^2 n^2 \rho}$, Algorithm 1 is $(\epsilon, \delta)$-DP with respect to the sensitive attributes for all data sets containing at least $\rho$-fraction of minority attributes. Further, if $\sigma_w^2 \geqslant \frac{32T \ln(1/\delta)}{\epsilon^2 n^2} \left( \frac{1}{\rho} + D^2 \right)$ and $\sigma_\theta^2 \geqslant \frac{64 L_\theta^2 D^2 \ln(1/\delta) T}{\epsilon^2 n^2 \rho} + \frac{32 D^4 L_\theta^2 l^2 T \ln(1/\delta)}{\epsilon^2 n^2}$, then Algorithm 1 is $(\epsilon, \delta)$-DP (with respect to all features) for all data sets containing at least $\rho$-fraction of minority attributes.*

See Appendix D for the proof. Next, we give a convergence guarantee for Algorithm 1:

**Theorem 3.4.** *Assume the loss function $\ell(\cdot, x, y)$ and $\mathcal{F}(x, \cdot)$ are Lipschitz continuous with Lipschitz gradient for all $(x, y)$, and $\widehat{P}_S(r) \geqslant \rho > 0 \ \forall \ r \in [k]$. In Algorithm 1, choose $\mathcal{W}$ to be*

*a sufficiently large ball that contains* $W^*(\theta) := \operatorname{argmax}_W \widehat{F}(\theta, W)$ *for every $\theta$ in some neighborhood of $\theta^* \in \operatorname{argmin}_\theta \max_W \widehat{F}(\theta, W)$. Then there exist algorithmic parameters such that the $(\epsilon, \delta)$-DP Algorithm 1 returns $\hat{\theta}_T$ with*

$$\mathbb{E}\|\nabla FERMI(\hat{\theta}_T)\|^2 = \mathcal{O}\left(\frac{\sqrt{\max(d_\theta, kl)\ln(1/\delta)}}{\epsilon n}\right),$$

*treating $D = \text{diameter}(\mathcal{W})$, $\lambda$, $\rho$, $l$, and the Lipschitz and smoothness parameters of $\ell$ and $\mathcal{F}$ as constants.*

Theorem 3.4 shows that Algorithm 1 finds an approximate stationary point of (FERMI obj.). Finding approximate stationary points is generally the best one can hope to do in polynomial time for non-convex optimization (Murty & Kabadi, 1985). The stationarity gap in Theorem 3.4 depends on the number of samples $n$ and model parameters $d_\theta$, the desired level of privacy $(\epsilon, \delta)$, and the number of labels $l$ and sensitive attributes $k$. For large-scale models (e.g. deep neural nets), we typically have $d_\theta \gg 1$ and $k, l = \mathcal{O}(1)$, so that the convergence rate of Algorithm 1 is essentially immune to the number of labels and sensitive attributes. In contrast, *no existing works with convergence guarantees are able to handle non-binary classification* ($l > 2$), *even with full batches* and a single binary sensitive attribute.

A few more remarks are in order. First, *the utility bound in Theorem 3.4 corresponds to DP for all of the features*. If DP is only required for the sensitive attributes, then using the smaller $\sigma_\theta^2, \sigma_w^2$ in Theorem 3.3 would improve the dependence on constants $D, l, L_\theta$ in the utility bound. Second, the choice of $\mathcal{W}$ in Theorem 3.4 implies that (4) is equivalent to $\min_\theta \max_{W \in \mathcal{W}} \widehat{F}(\theta, W)$, which is what our algorithm directly solves (c.f. (7)). Lastly, note that while we return a uniformly random iterate in Algorithm 1 for our theoretical convergence analysis, we recommend returning the last iterate $\theta_T$ in practice: our numerical experiments show strong performance of the last iterate.

In Theorem E.1 of Appendix E, we prove a result which is more general than Theorem 3.4. Theorem E.1 shows that noisy DP-SGDA converges to an approximate stationary point of *any* smooth nonconvex-strongly concave min-max optimization problem (not just (4)). We expect Theorem E.1 to be of general interest to the DP optimization community beyond its applications to DP fair learning, since it is the first DP convergence guarantee for nonconvex min-max optimization. We also give a bound on the iteration complexity $T$ in Appendix E.

The proof of Theorem E.1 involves a careful analysis of how the Gaussian noises propagate through the optimization trajectories of $\theta_t$ and $w_t$. Compared with DP non-convex *minimization* analyses (e.g. Wang et al. (2019); Hu et al. (2021); Ding et al. (2021b); Lowy et al. (2022b)), the two noises required to privatize the solution of the min-max problem we consider complicates the analysis and requires careful tuning of $\eta_\theta$ and $\eta_W$. Compared to existing analyses of DP min-max games in Boob & Guzmán (2021); Yang et al. (2022); Zhang et al. (2022), which assume that $f(\cdot, w)$ is *convex* or *PL*, dealing with *non-convexity* is a challenge that requires different optimization techniques.

## 4 NUMERICAL EXPERIMENTS

In this section, we evaluate the performance of our proposed approach (DP-FERMI) in terms of the fairness violation vs. test error for different privacy levels. We present our results in two parts: In Section 4.1, we assess the performance of our method in training logistic regression models on several benchmark tabular datasets. Since this is a standard setup that existing DP fairness algorithms can handle, we are able to compare our method against the state-of-the-art baselines. We carefully tuned the hyperparameters of all baselines for fair comparison. We find that *DP-FERMI consistently outperforms all state-of-the-art baselines across all data sets and all privacy levels*. These observations hold for both demographic parity and equalized odds fairness notions. To quantify the improvement of our results over the state-of-the-art baselines, we calculated the performance gain with respect to fairness violation (for fixed accuracy level) that our model yields over all the datasets. We obtained a performance gain of demographic parity that was 79.648 % better than Tran et al. (2021b) on average, and 65.89% better on median. The average performance gain of equalized odds was 96.65% while median percentage gain was 90.02%. In Section 4.2, we showcase the *scalability* of DP-FERMI by using it to train a deep convolutional neural network for classification on a large

image dataset. In Appendix F, we give detailed descriptions of the data sets, experimental setups and training procedure, along with additional results.

### 4.1 STANDARD BENCHMARK EXPERIMENTS: LOGISTIC REGRESSION ON TABULAR DATASETS

In the first set of experiments we train a logistic regression model using DP-FERMI (Algorithm 1) for demographic parity and a modified version of DP-FERMI (described in Appendix F) for equalized odds. We compare DP-FERMI against all applicable publicly available baselines in each expeiment.

#### 4.1.1 DEMOGRAPHIC PARITY

We use four benchmark tabular datasets: Adult Income, Retired Adult, Parkinsons, and Credit-Card dataset from the UCI machine learning repository (Dua & Graff (2017)). The predicted variables and sensitive attributes are both binary in these datasets. We analyze fairness-accuracy trade-offs with four different values of $\epsilon \in \{0.5, 1, 3, 9\}$ for each dataset. We compare against state-of-the-art algorithms proposed in Tran et al. (2021a) and (the demographic parity objective of) Tran et al. (2021b). The results displayed are averages over 15 trials (random seeds) for each value of $\epsilon$.

For the Adult dataset, the task is to predict *whether the income is greater than $50K or not* keeping *gender* as the sensitive attribute. The Retired Adult dataset is the same as the Adult dataset, but with updated data. We use the same output and sensitive attributes for both experiments. The results for Adult and Retired Adult are shown in Figs. 2 and 6 (in Appendix F.2). Compared to Tran et al. (2021a;b), DP-FERMI offers superior fairness-accuracy tradeoffs at every privacy ($\epsilon$) level.

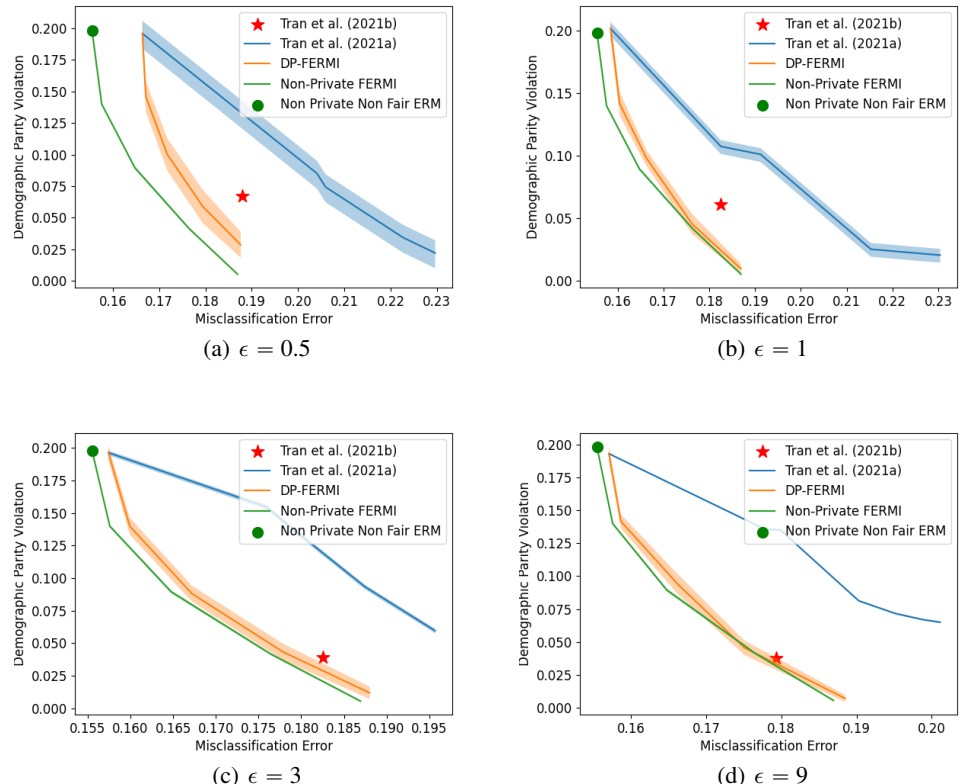

Figure 2: Private, Fair (Demographic Parity) logistic regression on Adult Dataset.

In the Parkinsons dataset, the task is to predict *whether the total UPDRS score of the patient is greater than the median or not* keeping *gender* as the sensitive attribute. Results for $\epsilon \in \{1, 3\}$ are shown in Fig. 3. See Fig. 8 in Appendix F for $\epsilon \in \{0.5, 9\}$. Our algorithm again outperforms the baselines Tran et al. (2021a;b) for all tested privacy levels.

In the Credit Card dataset , the task is to predict *whether the user will default payment the next month* keeping *gender* as the sensitive attribute. Results are shown in Fig. 7 in Appendix F.2. Once again, DP-FERMI provides the most favorable privacy-fairness-accuracy profile.

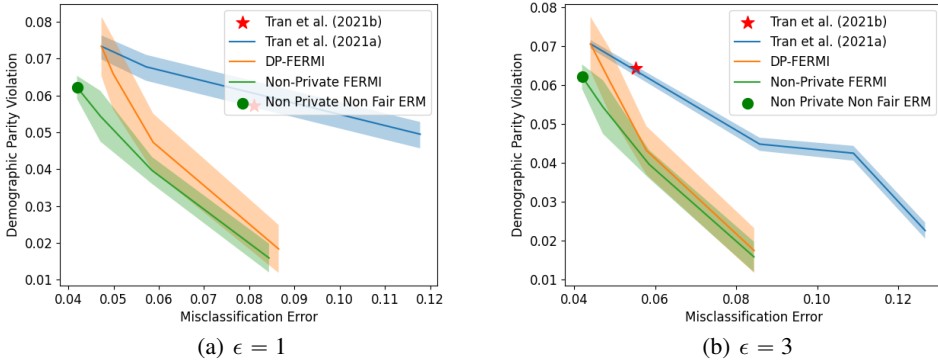

Figure 3: Private, Fair (Demogrpahic Parity) logistic regression on Parkinsons Dataset

### 4.1.2 EQUALIZED ODDS

Next, we consider the slightly modified version of Algorithm 1, which is designed to minimize the Equalized Odds violation by replacing the absolute probabilities in the objective with class conditional probabilities: see Appendix F.2.4 for details.

We considered the Credit Card and Adult datasets for these experiments, using the same sensitive attributes as mentioned above. Results for Credit Card are shown in Fig. 4. Adult results are given in Fig. 9 in Appendix F.2.4. Compared to Jagielski et al. (2019) and the equalized odds objective in Tran et al. (2021b), our *equalized odds variant of DP-FERMI outperforms these state-of-the-art baselines at every privacy level*.

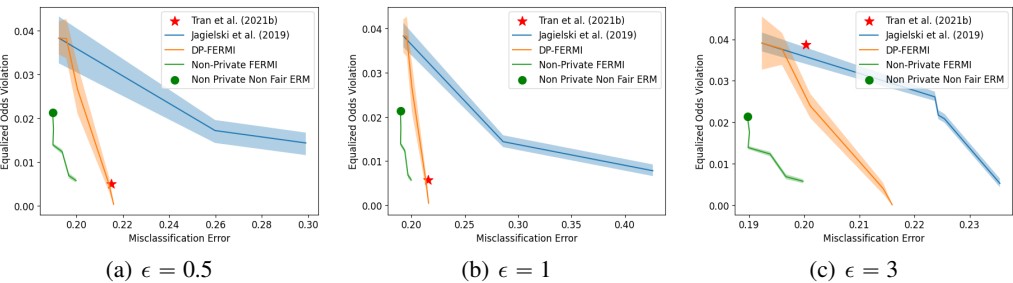

Figure 4: Private, Fair (Equalized Odds) logistic regression on Credit Card Dataset

### 4.2 LARGE-SCALE EXPERIMENT: DEEP CONVOLUTIONAL NEURAL NETWORK ON IMAGE DATASET

In our second set of experiments, we train a deep 9-layer VGG-like classifier (Simonyan & Zisserman, 2015) with $d \approx 1.6$ million parameters on the UTK-Face dataset (Zhang et al., 2017) using Algorithm 1. We classify the facial images into 9 age groups similar to the setup in Tran et al. (2022), while keeping *race* (containing 5 classes) as the sensitive attribute. See Appendix F.3 for more details. We analyze consider with four different privacy levels $\epsilon \in \{10, 25, 50, 100\}$. Compared to the tabular datasets, larger $\epsilon$ is needed to obtain stable results in the large-scale setting since the number of parameters $d$ is much larger and the cost of privacy increases with $d$ (see Theorem 3.4). Larger values of $\epsilon > 100$ were used in the baseline Jagielski et al. (2019) for smaller scale experiments.

The results in Fig. 5 empirically verify our main theoretical result: *DP-FERMI converges even for non-binary classification with small batch size and non-binary sensitive attributes.* We took Tran et al. (2021a;b) as our baselines and attempted to adapt them to this non-binary large-scale task. We observed that the baselines were very unstable while training and mostly gave degenerate results (predicting a single output irrespective of the input). By contrast, our method was able to obtain stable and meaningful tradeoff curves. Also, while Tran et al. (2022) reported results on UTK-Face, their code is not publicly available and we were unable to reproduce their results.

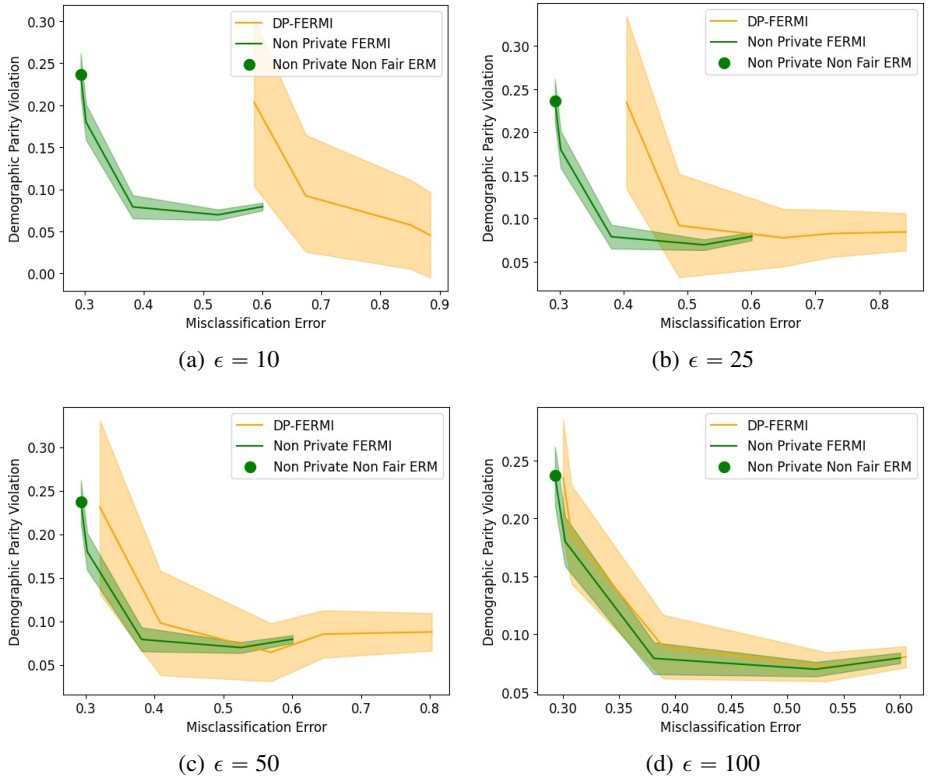

Figure 5: DP-FERMI on a Deep CNN for Image Classification on UTK-Face

## 5    CONCLUDING REMARKS

Motivated by pressing legal, ethical, and social considerations, we studied the challenging problem of learning fair models with differentially private demographic data. We observed that existing works suffer from a few crucial limitations that render their approaches impractical for large-scale problems. Specifically, existing approaches require full batches of data in each iteration (and/or exponential runtime) in order to provide convergence/accuracy guarantees. We addressed these limitations by deriving a DP stochastic optimization algorithm for fair learning, and rigorously proved the convergence of the proposed method. Our convergence guarantee holds even for non-binary classification (with any hypothesis class, even infinite VC dimension, c.f. Jagielski et al. (2019)) with multiple sensitive attributes and access to random minibatches of data in each iteration. Finally, we evaluated our method in extensive numerical experiments and found that it significantly outperforms the previous state-of-the-art models, in terms of fairness-accuracy tradeoff. The potential societal impacts of our work are discussed in Appendix G.

### ACKNOWLEDGMENTS

This work was supported in part with funding from the NSF CAREER award 2144985, from the YIP AFOSR award, from a gift from the USC-Meta Center for Research and Education in AI & Learning, and from a gift from the USC-Amazon Center on Secure & Trusted Machine Learning.

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

APPENDIX

## A   ADDITIONAL DISCUSSION OF RELATED WORK

The study of differentially private fair learning algorithms was initiated by Jagielski et al. (2019). Jagielski et al. (2019) considered equalized odds and proposed two DP algorithms: 1) an $\epsilon$-DP post-processing approach derived from Hardt et al. (2016a); and 2) an $(\epsilon, \delta)$-DP in-processing approach based on Agarwal et al. (2018). The major drawback of their post-processing approach is the unrealistic requirement that the algorithm have access to the sensitive attributes at test time, which Jagielski et al. (2019) admits "isn't feasible (or legal) in certain applications." Additionally, post-processing approaches are known to suffer from inferior fairness-accuracy tradeoffs compared with in-processing methods. While the in-processing method of Jagielski et al. (2019) does not require access to sensitive attributes at test time, it comes with a different set of disadvantages: 1) it is *limited to binary* classification; 2) its theoretical performance guarantees require the use of the *computationally inefficient* (i.e. exponential-time) exponential mechanism (McSherry & Talwar, 2007); 3) its theoretical performance guarantees require computations on the full training set and *do not permit mini-batch implementations*; 4) it requires the hypothesis class $\mathcal{H}$ to have finite VC dimension. In this work, we propose *the first algorithm that overcomes all of these pitfalls*: our algorithm is amenable to multi-way classification with multiple sensitive attributes, computationally efficient, and comes with convergence guarantees that hold even when mini-batches of $m < n$ samples are used in each iteration of training, and even when $\text{VC}(\mathcal{H}) = \infty$. Furthermore, our framework is flexible enough to accommodate many notions of group fairness besides equalized odds (e.g. demographic parity, accuracy parity).

Following Jagielski et al. (2019), several works have proposed other DP fair learning algorithms. *None of these works have managed to simultaneously address all the shortcomings* of the method of Jagielski et al. (2019). The work of Xu et al. (2019) proposed DP and fair binary logistic regression, but did not provide any theoretical convergence/performance guarantees. The work of Mozannar et al. (2020) combined aspects of both Hardt et al. (2016a) and Agarwal et al. (2018) in a two-step locally differentially private fairness algorithm. Their approach is *limited to binary classification*. Moreover, their algorithm requires $n/2$ samples in each iteration (of their in-processing step), making it *impractical for large-scale problems*. More recently, Tran et al. (2021b) devised another DP in-processing method based on lagrange duality, which covers non-binary classification problems. In a subsequent work, Tran et al. (2021a) studied the effect of DP on accuracy parity in ERM, and proposed using a regularizer to promote fairness. Finally, Tran et al. (2022) provided a semi-supervised fair "Private Aggregation of Teacher Ensembles" framework. A shortcoming of each of these three most recent works is their *lack of theoretical convergence or accuracy guarantees*. In another vein, some works have observed the disparate impact of privacy constraints on demographic subgroups (Bagdasaryan et al., 2019; Tran et al., 2021c).

## B   EQUALIZED ODDS VERSION OF ERMI

If equalized odds (Hardt et al., 2016b) is the desired fairness notion, then one should use the following variation of ERMI as a regularizer Lowy et al. (2022a):

$$\widehat{D}_R(\widehat{Y}; S|Y) := \mathbb{E}\left\{ \frac{\hat{p}_{\widehat{Y},S|Y}(\widehat{Y}, S|Y)}{\hat{p}_{\widehat{Y}|Y}(\widehat{Y}|Y)\hat{p}_{S|Y}(S|Y)} \right\} - 1$$

$$= \sum_{y=1}^{l} \sum_{j=1}^{l} \sum_{r=1}^{k} \frac{\hat{p}_{\widehat{Y},S|Y}(j, r|y)^2}{\hat{p}_{\widehat{Y}|Y}(j|y)\hat{p}_{S|Y}(r|y)} \hat{p}_Y(y) - 1. \quad (5)$$

Here $\hat{p}_{\widehat{Y},S|Y}$ denotes the empirical joint distribution of the predictions and sensitive attributes $(\widehat{Y}, S)$ conditional on the true labels $Y$. In particular, if $D_R(\widehat{Y}; S|Y) = 0$, then $\widehat{Y}$ and $S$ are conditionally independent given $Y$ (i.e. equalized odds is satisfied).

## C COMPLETE VERSION OF THEOREM 3.2

Let $\widehat{\mathbf{y}}(x_i; \theta) \in \{0,1\}^l$ and $\mathbf{s}_i \in \{0,1\}^k$ be the one-hot encodings of $\widehat{y}(x_i, \theta)$ and $s_i$, respectively: i.e., $\widehat{\mathbf{y}}_j(x_i; \theta) = \mathbb{1}_{\{\widehat{y}(x_i,\theta)=j\}}$ and $\mathbf{s}_{i,r} = \mathbb{1}_{\{s_i=r\}}$ for $j \in [l], r \in [k]$. Also, denote $\widehat{P}_s = \mathrm{diag}(\widehat{p}_S(1), \ldots, \widehat{p}_S(k))$, where $\widehat{p}_S(r) := \frac{1}{n} \sum_{i=1}^n \mathbb{1}_{\{s_i=r\}} \geqslant \rho > 0$ is the empirical probability of attribute $r$ ($r \in [k]$). Then we have the following re-formulation of (FERMI obj.) as a min-max problem:

**Theorem C.1** (Lowy et al. (2022a)). (FERMI obj.) *is equivalent to*

$$\min_\theta \max_{W \in \mathbb{R}^{k \times l}} \left\{ \widehat{F}(\theta, W) := \widehat{\mathcal{L}}(\theta) + \lambda \frac{1}{n} \sum_{i=1}^n \widehat{\psi}_i(\theta, W) \right\}, \tag{6}$$

*where*

$$\widehat{\psi}_i(\theta, W) := - \mathrm{Tr}(W \mathbb{E}[\widehat{\mathbf{y}}(x_i, \theta)\widehat{\mathbf{y}}(x_i, \theta)^T | x_i] W^T)$$
$$+ 2 \mathrm{Tr}(W \mathbb{E}[\widehat{\mathbf{y}}(x_i; \theta)\mathbf{s}_i^T | x_i, \mathbf{s}_i] \widehat{P}_s^{-1/2}) - 1,$$

$\mathbb{E}[\widehat{\mathbf{y}}(x_i; \theta)\widehat{\mathbf{y}}(x_i; \theta)^T | x_i] = \mathrm{diag}(\mathcal{F}_1(x_i, \theta), \ldots, \mathcal{F}_l(x_i, \theta))$, *and* $\mathbb{E}[\widehat{\mathbf{y}}(x_i; \theta)\mathbf{s}_i^T | x_i, \mathbf{s}_i]$ *is a* $k \times l$ *matrix with* $\mathbb{E}[\widehat{\mathbf{y}}(x_i; \theta)\mathbf{s}_i^T | x_i, \mathbf{s}_i]_{r,j} = \mathbf{s}_{i,r} \mathcal{F}_j(x_i, \theta)$.

Strong concavity of $\widehat{\psi}_i$ is shown in Lowy et al. (2022a).

## D DP-FERMI ALGORITHM: PRIVACY

We begin with a routine calculation of the derivatives of $\widehat{\psi}_i$, which follows by elementary matrix calculus:

**Lemma D.1.** *Let* $\widehat{\psi}_i(\theta, W) = - \mathrm{Tr}(W \mathbb{E}[\widehat{\mathbf{y}}(x_i, \theta)\widehat{\mathbf{y}}(x_i, \theta)^T | x_i] W^T) + 2 \mathrm{Tr}(W \mathbb{E}[\widehat{\mathbf{y}}(x_i; \theta)\mathbf{s}_i^T | x_i, \mathbf{s}_i] \widehat{P}_s^{-1/2}) - 1$, *where* $\mathbb{E}[\widehat{\mathbf{y}}(x_i; \theta)\widehat{\mathbf{y}}(x_i; \theta)^T | x_i] = \mathrm{diag}(\mathcal{F}_1(x_i, \theta), \ldots, \mathcal{F}_l(x_i, \theta))$ *and* $\mathbb{E}[\widehat{\mathbf{y}}(x_i; \theta)\mathbf{s}_i^T | x_i, \mathbf{s}_i]$ *is a* $k \times l$ *matrix with* $\mathbb{E}[\widehat{\mathbf{y}}(x_i; \theta)\mathbf{s}_i^T | x_i, \mathbf{s}_i]_{r,j} = \mathbf{s}_{i,r} \mathcal{F}_j(x_i, \theta)$. *Then,*

$$\nabla_\theta \widehat{\psi}_i(\theta, W) = -\nabla_\theta \mathrm{vec}(\mathbb{E}[\widehat{\mathbf{y}}(x_i, \theta)\widehat{\mathbf{y}}(x_i, \theta)^T | x_i])^T \mathrm{vec}(W^T W) + 2\nabla_\theta \mathrm{vec}(\mathbb{E}[\mathbf{s}_i \widehat{\mathbf{y}}(x_i, \theta)^T | x_i, s_i]) \mathrm{vec}\left(W^T \left(\widehat{P}_S\right)^{-1/2}\right)$$

*and*

$$\nabla_w \widehat{\psi}_i(\theta, W) = -2W \mathbb{E}[\widehat{\mathbf{y}}(x_i, \theta)\widehat{\mathbf{y}}(x_i, \theta)^T | x_i] + 2\widehat{P}_S^{-1/2} \mathbb{E}[\mathbf{s}_i \widehat{\mathbf{y}}(x_i, \theta)^T | x_i, s_i].$$

Using Theorem D.1, we can prove that Algorithm 1 is DP:

**Theorem D.2** (Re-statement of Theorem 3.3). *Let* $\epsilon \leqslant 2\ln(1/\delta)$, $\delta \in (0,1)$, *and* $T \geqslant \left(n \frac{\sqrt{\epsilon}}{2m}\right)^2$. *Assume* $\mathcal{F}(\cdot, x)$ *is* $L_\theta$-Lipschitz for all $x$, and $|(W_t)_{r,j}| \leqslant D$ for all $t \in [T], r \in [k], j \in [l]$. *Then, for* $\sigma_w^2 \geqslant \frac{16T\ln(1/\delta)}{\epsilon^2 n^2 \rho}$ *and* $\sigma_\theta^2 \geqslant \frac{16L_\theta^2 D^2 \ln(1/\delta)T}{\epsilon^2 n^2 \rho}$, *Algorithm 1 is* $(\epsilon, \delta)$-*DP with respect to the sensitive attributes for all data sets containing at least* $\rho$-*fraction of minority attributes. Further, if* $\sigma_w^2 \geqslant \frac{32T\ln(1/\delta)}{\epsilon^2 n^2} \left(\frac{1}{\rho} + D^2\right)$ *and* $\sigma_\theta^2 \geqslant \frac{64L_\theta^2 D^2 \ln(1/\delta)T}{\epsilon^2 n^2 \rho} + \frac{32D^4 L_\theta^2 l^2 T \ln(1/\delta)}{\epsilon^2 n^2}$, *then Algorithm 1 is* $(\epsilon, \delta)$-*DP (with respect to all features) for all data sets containing at least* $\rho$-*fraction of minority attributes.*

*Proof.* First consider the case in which only the sensitive attributes are private. By the moments accountant Theorem 1 in Abadi et al. (2016), it suffices to bound the sensitivity of the gradient updates by $\Delta_\theta^2 \leqslant \frac{8D^2 L_\theta^2}{m^2 \rho}$ and $\Delta_w^2 \leqslant \frac{8}{m^2 \rho}$. Here

$$\Delta_\theta^2 = \sup_{Z \sim Z', \theta, W} \left\| \frac{1}{m} \sum_{i \in B_t} \left[ \nabla_\theta \widehat{\psi}(\theta, W; z_i) - \nabla_\theta \widehat{\psi}(\theta, W; z_i') \right] \right\|^2$$

and $Z \sim Z'$ means that $Z$ and $Z'$ are two data sets (both with $\rho$-fraction of minority attributes) that differ in exactly one person's sensitive attributes: i.e. $s_i \neq s_i'$ for some unique $i \in [n]$, but $z_j = z_j'$ for all $j \neq i$ and $(x_i, y_i) = (x_i', y_i')$. Likewise,

$$\Delta_w^2 = \sup_{Z \sim Z', \theta, W} \left\| \frac{1}{m} \sum_{i \in B_t} \left[ \nabla_w \widehat{\psi}(\theta, W; z_i) - \nabla_w \widehat{\psi}(\theta, W; z_i') \right] \right\|^2.$$

Now, by Theorem D.1,

$$\nabla_\theta \widehat{\psi}_i(\theta, W) = -\nabla_\theta \operatorname{vec}(\mathbb{E}[\widehat{\mathbf{y}}(x_i, \theta)\widehat{\mathbf{y}}(x_i, \theta)^T | x_i])^T \operatorname{vec}(W^T W)$$
$$+ 2\nabla_\theta \operatorname{vec}(\mathbb{E}[\mathbf{s}_i\widehat{\mathbf{y}}(x_i, \theta)^T | x_i, s_i]) \operatorname{vec}\left( W^T \left( \widehat{P}_S \right)^{-1/2} \right),$$

and notice that only the second term depends on $S$. Therefore, we can bound the $\ell_2$-sensitivity of the $\theta$-gradient updates by:

$$\Delta_\theta^2 = \sup_{Z \sim Z', W, \theta} \left\| \frac{1}{m} \sum_{i=1}^m 2\nabla_\theta \operatorname{vec}(\mathbb{E}[\mathbf{s}_i\widehat{\mathbf{y}}(x_i, \theta)^T | x_i, s_i]) \operatorname{vec}\left( W^T \left( \widehat{P}_S \right)^{-1/2} \right) \right.$$
$$\left. - 2\nabla_\theta \operatorname{vec}(\mathbb{E}[\mathbf{s}_i'\widehat{\mathbf{y}}(x_i, \theta)^T | x_i, s_i']) \operatorname{vec}\left( W^T \left( \widehat{P}_{S'} \right)^{-1/2} \right) \right\|^2$$

$$\leqslant \frac{4}{m^2} \sup_{x, \mathbf{s}_i, \mathbf{s}_i', W, \theta} \left[ \sum_{r=1}^k \sum_{j=1}^l \|\nabla_\theta \mathcal{F}_j(\theta, x)\|^2 W_{r,j}^2 \left( \frac{s_{i,r}}{\sqrt{\widehat{P}_S(r)}} - \frac{s_{i,r}'}{\sqrt{\widehat{P}_{S'}(r)}} \right)^2 \right]$$

$$\leqslant \frac{8}{\rho m^2} \sup_{x, W, \theta} \left( \sum_{j=1}^l \|\nabla_\theta \mathcal{F}_j(\theta, x)\|^2 W_{r,j}^2 \right)$$

$$\leqslant \frac{8 D^2 L_\theta^2}{\rho m^2},$$

using Lipschitz continuity of $\mathcal{F}(\cdot, x)$, the assumption that $\mathcal{W}$ has diameter bounded by $D$, the assumption that the data sets have at least $\rho$-fraction of sensitive attribute $r$ for all $r \in [k]$. Similarly, for the $W$-gradients, we have

$$\nabla_w \widehat{\psi}_i(\theta, W) = -2W\mathbb{E}[\widehat{\mathbf{y}}(x_i, \theta)\widehat{\mathbf{y}}(x_i, \theta)^T | x_i] + 2\widehat{P}_S^{-1/2}\mathbb{E}[\mathbf{s}_i\widehat{\mathbf{y}}(x_i, \theta)^T | x_i, s_i]$$

by Theorem D.1. Hence

$$\Delta_W^2 = \sup_{\theta, W, \mathbf{s}_i, \mathbf{s}_i'} \frac{4}{m^2} \left\| - W \operatorname{diag}(\mathcal{F}_1(\theta, x_i), \ldots, \mathcal{F}_l(\theta, x_i)) + \widehat{P}_S^{-1/2}\mathbb{E}[\mathbf{s}_i\widehat{\mathbf{y}}_i(x_i; \theta_t)^T | x_i, s_i] \right.$$

$$\left. + W \operatorname{diag}(\mathcal{F}_1(\theta, x_i), \ldots, \mathcal{F}_l(\theta, x_i)) - \widehat{P}_{S'}^{-1/2}\mathbb{E}[\mathbf{s}_i'\widehat{\mathbf{y}}_i(x_i; \theta_t)^T | x_i, s_i'] \right\|^2$$

$$\leqslant \frac{4}{m^2} \sup_{\theta, W, \mathbf{s}_i, \mathbf{s}_i'} \sum_{j=1}^l \mathcal{F}_j(\theta, x_i)^2 \sum_{r=1}^k \left( \frac{s_{i,r}}{\sqrt{\widehat{P}_S(r)}} - \frac{s_{i,r}'}{\sqrt{\widehat{P}_{S'}(r)}} \right)^2$$

$$\leqslant \frac{8}{m^2\rho},$$

since $\sum_{j=1}^l \mathcal{F}_j(\theta, x_i)^2 \leqslant \sum_{j=1}^l \mathcal{F}_j(\theta, x_i) = 1$. This establishes the desired privacy guarantee with respect to sensitive attributes for Algorithm 1.

Now consider the case in which all features are private. We aim to bound the sensitivities of the gradient updates to changes in a single sample $z_i = (s_i, x_i, y_i)$. Denote these new sensitivities by

$$\tilde{\Delta}_\theta = \sup_{Z \sim Z', \theta, W} \left\| \frac{1}{m} \sum_{i \in B_t} \left[ \nabla_\theta \widehat{\psi}(\theta, W; z_i) - \nabla_\theta \widehat{\psi}(\theta, W; z_i') \right] \right\|,$$

where we now write $Z \sim Z'$ to mean that $Z$ and $Z'$ are two data sets (both with $\rho$-fraction of minority attributes) that differ in exactly one person's (sensitive and non-sensitive) data: i.e. $z_i \neq z'_i$ for some unique $i \in [n]$. Likewise,

$$\tilde{\Delta}_W = \sup_{Z \sim Z', \theta, W} \left\| \frac{1}{m} \sum_{i \in B_t} \left[ \nabla_w \widehat{\psi}(\theta, W; z_i) - \nabla_w \widehat{\psi}(\theta, W; z'_i) \right] \right\|.$$

Then

$$\tilde{\Delta}_\theta = \frac{1}{m} \sup_{z_i, z'_i, \theta, W, S \sim S'} \left\| - \nabla_\theta \operatorname{vec}(\mathbb{E}[\widehat{\mathbf{y}}(x_i, \theta) \widehat{\mathbf{y}}(x_i, \theta)^T | x_i])^T \operatorname{vec}(W^T W) + 2 \nabla_\theta \operatorname{vec}(\mathbb{E}[\mathbf{s}_i \widehat{\mathbf{y}}(x_i, \theta)^T | x_i, s_i]) \right.$$

$$\operatorname{vec} \left( W^T \left( \widehat{P}_S \right)^{-1/2} \right) + \nabla_\theta \operatorname{vec}(\mathbb{E}[\widehat{\mathbf{y}}(x'_i, \theta) \widehat{\mathbf{y}}(x'_i, \theta)^T | x'_i])^T \operatorname{vec}(W^T W)$$

$$\left. - 2 \nabla_\theta \operatorname{vec}(\mathbb{E}[\mathbf{s}'_i \widehat{\mathbf{y}}(x'_i, \theta)^T | x'_i, s'_i]) \operatorname{vec} \left( W^T \left( \widehat{P}_{S'} \right)^{-1/2} \right) \right\|$$

$$\leqslant \frac{2 L_\theta l D}{m} + \Delta_\theta.$$

Thus, $\tilde{\Delta}_\theta^2 \leqslant \frac{4 L_\theta^2 l^2 D^2}{m^2} + 2 \Delta_\theta^2$. Therefore, by the moments accountant, the collection of all $\theta_t$ updates in Algorithm 1 is $(\epsilon, \delta)$-DP if $\sigma_\theta^2 \geqslant \frac{32 D^2 L_\theta^2 T \ln(1/\delta)}{\rho \epsilon^2 n^2} + \frac{8 D^2 L_\theta^2 l^2 T \ln(1/\delta)}{\epsilon^2 n^2} = \frac{8 L_\theta^2 D^2 T \ln(1/\delta)}{\epsilon^2 n^2} \left( \frac{4}{\rho} + l^2 \right)$.

Next, we bound the sensitivity $\tilde{\Delta}_W$ of the $W$-gradient updates. We have

$$\tilde{\Delta}_W^2 = \sup_{\theta, W, z_i, z'_i} \frac{4}{m^2} \left\| - W \operatorname{diag}(\mathcal{F}_1(\theta, x_i), \ldots, \mathcal{F}_l(\theta, x_i)) + \widehat{P}_S^{-1/2} \mathbb{E}[\mathbf{s}_i \widehat{\mathbf{y}}_i(x_i; \theta_t)^T | x_i, s_i] \right.$$

$$\left. + W \operatorname{diag}(\mathcal{F}_1(\theta, x'_i), \ldots, \mathcal{F}_l(\theta, x'_i)) - \widehat{P}_{S'}^{-1/2} \mathbb{E}[\mathbf{s}'_i \widehat{\mathbf{y}}_i^T (x'_i; \theta_t) | x'_i, s'_i] \right\|^2$$

$$\leqslant 2 \Delta_W^2 + \frac{8}{m^2} \sup_{\theta, W, x_i, x'_i} \left\| W \operatorname{diag}(\mathcal{F}_1(\theta, x_i) - \mathcal{F}_1(\theta, x'_i), \ldots, \mathcal{F}_l(\theta, x_i) - \mathcal{F}_l(\theta, x'_i)) \right\|^2$$

$$\leqslant 2 \Delta_W^2 + \frac{16 D^2}{m^2} \sup_{\theta, x_i} \sum_{j=1}^{l} \mathcal{F}_j(\theta, x_i)^2$$

$$\leqslant 2 \Delta_W^2 + \frac{16 D^2}{m^2}.$$

Therefore, by the moments accountant, the collection of all $W_t$ updates in Algorithm 1 is $(\epsilon, \delta)$-DP if $\sigma_w^2 \geqslant \frac{32 T \ln(1/\delta)}{\epsilon^2 n^2} \left( \frac{1}{\rho} + D^2 \right)$. This completes the proof. □

# E  DP-FERMI ALGORITHM: UTILITY

To prove Theorem 3.4, we will first derive a more general result. Namely, in Appendix E.1, we will provide a precise upper bound on the stationarity gap of noisy DP stochastic gradient descent ascent (DP-SGDA).

## E.1  NOISY DP-SGDA FOR NONCONVEX-STRONGLY CONCAVE MIN-MAX PROBLEMS

Consider a generic (smooth) nonconvex-strongly concave min-max ERM problem:

$$\min_{\theta \in \mathbb{R}^{d_\theta}} \max_{w \in \mathcal{W}} \left\{ F(\theta, w) := \frac{1}{n} \sum_{i=1}^{n} f(\theta, w; z_i) \right\}, \tag{7}$$

where $f(\theta, \cdot; z)$ is $\mu$-strongly concave[3] for all $\theta, z$ but $f(\cdot, w; z)$ is potentially non-convex. We

---

[3]We say a differentiable function $g$ is $\mu$-strongly concave if $g(\alpha) + \langle \nabla g(\alpha), \alpha' - \alpha \rangle - \frac{\mu}{2} \|\alpha - \alpha'\|^2 \geqslant g(\alpha')$ for all $\alpha, \alpha'$.

---

**Algorithm 2** Noisy Differentially Private Stochastic Gradient Descent-Ascent (DP-SGDA)

---

1: **Input**: data $Z$, $\theta_0 \in \mathbb{R}^{d_\theta}$, $w_0 \in \mathcal{W}$, step-sizes $(\eta_\theta, \eta_w)$, privacy noise parameters $\sigma_\theta, \sigma_w$, batch size $m$, iteration number $T \geqslant 1$.
2: **for** $t = 0, 1, \ldots, T-1$ **do**
3:     Draw a batch of data points $\{z_i\}_{i=1}^m$ uniformly at random from $Z$.
4:     Update $\theta_{t+1} \leftarrow \theta_t - \eta_\theta \left(\frac{1}{m} \sum_{i=1}^m \nabla_\theta f(\theta_t, w_t; z_i) + u_t\right)$, where $u_t \sim \mathcal{N}(0, \sigma_\theta^2 \mathbf{I}_{d_\theta})$ and
    $w_{t+1} \leftarrow \Pi_{\mathcal{W}} \left[w_t + \eta_w \left(\frac{1}{m} \sum_{i=1}^m \nabla_w f(\theta_t, w_t; z_i) + v_t\right)\right]$, where $v_t \sim \mathcal{N}(0, \sigma_w^2 \mathbf{I}_{d_w})$.
5: **end for**
6: Draw $\hat{\theta}_T$ uniformly at random from $\{\theta_t\}_{t=1}^T$.
7: **Return:** $\hat{\theta}_T$

---

propose Noisy DP-SGDA[4] (Algorithm 2) for privately solving (7), which is a noisy DP variation of two-timescale SGDA (Lin et al., 2020). Now, we provide *the first theoretical convergence guarantee for DP non-convex min-max optimization*:

**Theorem E.1** (Privacy and Utility of Algorithm 2, Informal Version)**.** *Let $\epsilon \leqslant 2\ln(1/\delta)$, $\delta \in (0,1)$. Assume:* $f(\cdot, w; z)$ *is $L_\theta$-Lipschitz[5] and* $f(\theta, \cdot; z)$ *is $L_w$-Lipschitz for all $\theta, w, z$; and $\mathcal{W} \subset \mathbb{R}^{d_w}$ is a convex, compact set. Denote $\Phi(\theta) = \max_{w \in \mathcal{W}} F(\theta, w)$. Choose $\sigma_w^2 = \frac{8TL_w^2 \ln(1/\delta)}{\epsilon^2 n^2}$, $\sigma_\theta^2 = \frac{8TL_\theta^2 \ln(1/\delta)}{\epsilon^2 n^2}$, and $T \geqslant \left(n\frac{\sqrt{\epsilon}}{2m}\right)^2$. Then, Algorithm 2 is $(\epsilon, \delta)$-DP. Further, if $f(\cdot, \cdot; z)$ has Lipschitz gradients and $f(\theta, \cdot; z)$ is strongly concave, then $\exists\, T, \eta_\theta, \eta_w$ such that*

$$\mathbb{E}\|\nabla\Phi(\hat{\theta}_T)\|^2 = \mathcal{O}\left(\frac{\sqrt{d\ln(1/\delta)}}{\epsilon n}\right),$$

*where $d = \max(d_\theta, d_w)$. (The expectation is solely over the algorithm.)*

In our DP fair learning application, $f(\theta, W; z_i) = \ell(\theta, x_i, y_i) + \lambda\widehat{\psi}_i(\theta, W)$ and the strong concavity assumption on $f$ in Theorem E.1 is automatically satisfied, by Lowy et al. (2022a). The Lipschitz and smoothness assumptions on $f$ are standard in optimization literature and are satisfied for loss functions that are typically used in pracdtice. In our application to DP-FERMI, these assumptions hold as long as the loss function $\ell$ and $\mathcal{F}$ are Lipschitz continuous with Lipschitz gradients. Our next goal is to prove (the precise, scale-invariant version of) Theorem E.1. To that end, we require the following notation.

**Notation and Assumptions:** Let $f : \mathbb{R}^{d_\theta} \times \mathbb{R}^{d_w} \times \mathcal{Z} \to \mathbb{R}$, and $F(\theta, w) = \frac{1}{n} \sum_{i=1}^n f(\theta, w; z_i)$ for fixed training data $Z = (z_1, \cdots, z_n) \in \mathcal{Z}^n$. Let $\mathcal{W} \subset \mathbb{R}^{d_w}$ be a convex, compact set. For any $\theta \in \mathbb{R}^{d_\theta}$, denote $w^*(\theta) \in \text{argmax}_{w \in \mathcal{W}} F(\theta, w)$ and $\widehat{\Phi}(\theta) = \max_{w \in \mathcal{W}} F(\theta, w)$. Let $\Delta_\Phi = \widehat{\Phi}(\theta_0) - \inf_\theta \widehat{\Phi}_Z(\theta)$. Recall that a function $h$ is $\beta$-smooth if its derivative $\nabla h$ is $\beta$-Lipschitz. We write $a \lesssim b$ if there is an absolute constant $C > 0$ such that $a \leqslant Cb$.

**Assumption E.2.**     1. $f(\cdot, w; z)$ is $L_\theta$-Lipschitz and $\beta_\theta$-smooth for all $w \in \mathcal{W}, z \in \mathcal{Z}$.

    2. $f(\theta, \cdot; z)$ is $L_w$-Lipschitz, $\beta_w$-smooth, and $\mu$-strongly concave on $\mathcal{W}$ for all $\theta \in \mathbb{R}^{d_\theta}$, $z \in \mathcal{Z}$.

    3. $\|\nabla_w f(\theta, w; z) - \nabla_w f(\theta', w; z)\| \leqslant \beta_{\theta w}\|\theta - \theta'\|$ and $\|\nabla_\theta f(\theta, w; z) - \nabla_\theta f(\theta, w'; z)\| \leqslant \beta_{\theta w}\|w - w'\|$ for all $\theta, \theta', w, w', z$.

    4. $\mathcal{W}$ has $\ell_2$ diameter bounded by $D \geqslant 0$.

    5. $\nabla_w F(\theta, w^*(\theta)) = 0$ for all $\theta$, where $w^*(\theta)$ denotes the unconstrained global minimizer of $F(\theta, \cdot)$.

The first four assumptions are standard in (DP and min-max) optimization. The fifth assumption means that $\mathcal{W}$ contains the *unconstrained* global minimizer $w^*(\theta)$ of $F(\theta, \cdot)$ for all $\theta$. Hence (7) is equivalent to

$$\min_{\theta \in \mathbb{R}^{d_\theta}} \max_{w \in \mathbb{R}^{d_w}} F(\theta, w).$$

---

[4]DP-SGDA was also used in Yang et al. (2022) for *convex* and *PL* min-max problems.
[5]We say function $g$ is $L$-Lipschitz if $\|g(\alpha) - g(\alpha')\| \leqslant L\|\alpha - \alpha'\|$ for all $\alpha, \alpha'$.

This assumption is not actually necessary for our convergence result to hold, but we will need it when we *apply* our results to the DP fairness problem. Moreover, it simplifies the proof of our convergence result. We refer to problems of the form (7) that satisfy Theorem E.2 as "(smooth) nonconvex-strongly concave min-max." We denote $\kappa_w := \frac{\beta_w}{\mu}$ and $\kappa_{\theta w} := \frac{\beta_{\theta w}}{\mu}$.

We can now provide the complete, precise version of Theorem E.1:

**Theorem E.3** (Privacy and Utility of Algorithm 2, Formal Version). *Let* $\epsilon \leqslant 2\ln(1/\delta)$, $\delta \in (0,1)$. *Grant Theorem E.2. Choose* $\sigma_w^2 = \frac{8TL_w^2\ln(1/\delta)}{\epsilon^2 n^2}$, $\sigma_\theta^2 = \frac{8TL_\theta^2\ln(1/\delta)}{\epsilon^2 n^2}$, *and* $T \geqslant \left(n\frac{\sqrt{\epsilon}}{2m}\right)^2$. *Then Algorithm 2 is* $(\epsilon,\delta)$-*DP. Further, if we choose* $\eta_\theta = \frac{1}{16\kappa_w(\beta_\theta + \beta_{\theta w}\kappa_{\theta w})}$, $\eta_w = \frac{1}{\beta_w}$, *and* $T \approx \sqrt{\kappa_w[\Delta_\Phi(\beta_\theta + \beta_{\theta w}\kappa_{\theta w}) + \beta_{\theta w}^2 D^2]}\epsilon n\min\left(\frac{1}{L_\theta\sqrt{d_\theta}}, \frac{\beta_w}{\beta_{\theta w}L_w\sqrt{\kappa_w d_w}}\right)$, *then*

$$\mathbb{E}\|\nabla\Phi(\hat{\theta}_T)\|^2 \lesssim \sqrt{\Delta_\Phi(\beta_\theta + \beta_{\theta w}\kappa_{\theta w})\kappa_w + \kappa_w\beta_{\theta w}^2 D^2}\left[\frac{L_\theta\sqrt{d_\theta\ln(1/\delta)}}{\epsilon n} + \left(\frac{\beta_{\theta w}\sqrt{\kappa_w}}{\beta_w}\right)\frac{L_w\sqrt{d_w\ln(1/\delta)}}{\epsilon n}\right]$$

$$+ \frac{\mathbb{1}_{\{m<n\}}}{m}\left(L_\theta^2 + \frac{\kappa_w\beta_{\theta w}^2 L_w^2}{\beta_w^2}\right).$$

*In particular, if* $m \geqslant \min\left(\frac{\epsilon n L_\theta}{\sqrt{d_\theta\kappa_w[\Delta_\Phi(\beta_\theta + \beta_{\theta w}\kappa_{\theta w}) + \beta_{\theta w}^2 D^2]}}, \frac{\epsilon n L_w\sqrt{\kappa_w}}{\beta_{\theta w}\beta_w\sqrt{d_w\kappa_w[\Delta_\Phi(\beta_\theta + \beta_{\theta w}\kappa_{\theta w}) + \beta_{\theta w}^2 D^2]}}\right)$, *then*

$$\mathbb{E}\|\nabla\Phi(\hat{\theta}_T)\|^2 \lesssim \sqrt{\kappa_w[\Delta_\Phi(\beta_\theta + \beta_{\theta w}\kappa_{\theta w}) + \beta_{\theta w}^2 D^2]}\left(\frac{\sqrt{\ln(1/\delta)}}{\epsilon n}\right)\left(L_\theta\sqrt{d_\theta} + \left(\frac{\beta_{\theta w}\sqrt{\kappa_w}}{\beta_w}\right)L_w\sqrt{d_w}\right).$$

The proof of Theorem E.3 will require several technical lemmas. These technical lemmas, in turn, require some preliminary lemmas, which we present below.

We begin with a refinement of Lemma 4.3 from Lin et al. (2020):

**Lemma E.4.** *Grant Theorem E.2. Then* $\Phi$ *is* $2(\beta_\theta + \beta_{\theta w}\kappa_{\theta w})$-*smooth with* $\nabla\Phi(\theta) = \nabla_\theta F(\theta, w^*(\theta))$, *and* $w^*(\cdot)$ *is* $\kappa_w$-*Lipschitz.*

*Proof.* The proof follows almost exactly as in the proof of Lemma 4.3 of Lin et al. (2020), using Danskin's theorem, but we carefully track the different smoothness parameters with respect to $w$ and $\theta$ (and their units) to obtain the more precise result. □

**Lemma E.5** (Lei et al. (2017)). *Let* $\{a_l\}_{l\in[n]}$ *be an arbitrary collection of vectors such that* $\sum_{l=1}^n a_l = 0$. *Further, let* $\mathcal{S}$ *be a uniformly random subset of* $[n]$ *of size* $m$. *Then,*

$$\mathbb{E}\left\|\frac{1}{m}\sum_{l\in\mathcal{S}}a_l\right\|^2 = \frac{n-m}{(n-1)m}\frac{1}{n}\sum_{l=1}^n\|a_l\|^2 \leqslant \frac{\mathbb{1}_{\{m<n\}}}{m\,n}\sum_{l=1}^n\|a_l\|^2.$$

**Lemma E.6** (Co-coercivity of the gradient). *For any* $\beta$-*smooth and convex function* $g$, *we have*

$$\|\nabla g(a) - \nabla g(b)\|^2 \leqslant 2\beta(g(a) - g(b) - \langle g(b), a - b\rangle),$$

*for all* $a, b \in domain(g)$.

Having recalled the necessary preliminaries, we now provide the novel technical ingredients that we'll need for the proof of Theorem E.3. The next lemma quantifies the progress made in minimizing $\Phi$ from a single step of noisy stochastic gradient descent in $\theta$ (i.e. line 4 of Algorithm 2):

**Lemma E.7.** *For all* $t \in [T]$, *the iterates of Algorithm 2 satisfy*

$$\mathbb{E}\Phi(\theta_t) \leqslant \Phi(\theta_{t-1}) - \left(\frac{\eta_\theta}{2} - 2(\beta_\theta + \beta_{\theta w}\kappa_{\theta w})\eta_\theta^2\right)\mathbb{E}\|\nabla\Phi(\theta_{t-1})\|^2$$

$$+ \left(\frac{\eta_\theta}{2} + 2\eta_\theta^2(\beta_\theta + \beta_{\theta w}\kappa_{\theta w})\mathbb{E}\|\nabla\Phi(\theta_{t-1}) - \nabla_\theta F(\theta_{t-1}, w_{t-1})\|^2\right) + (\beta_\theta\beta_{\theta w}\kappa_{\theta w})\eta_\theta^2\left(d_\theta\sigma_\theta^2 + \frac{4L_\theta^2}{m}\mathbb{1}_{\{m<n\}}\right),$$

*conditional on* $\theta_{t-1}, w_{t-1}$.

*Proof.* Let us denote $\widetilde{g} := \frac{1}{m}\sum_{i=1}^{m}\nabla_\theta f(\theta_{t-1}, w_{t-1}; z_i) + u_{t-1} := g + u_{t-1}$, so $\theta_t = \theta_{t-1} - \eta_\theta \widetilde{g}$. First condition on the randomness due to sampling and Gaussian noise addition. By smoothness of $\Phi$ (see Theorem E.4), we have

$$\Phi(\theta_t) \leqslant \Phi(\theta_{t-1}) - \eta_\theta \langle \widetilde{g}, \nabla\Phi(\theta_{t-1})\rangle + (\beta_\theta + \beta_{\theta w}\kappa_{\theta w})\eta_\theta^2 \|\widetilde{g}\|^2$$
$$= \Phi(\theta_{t-1}) - \eta_\theta \|\nabla\Phi(\theta_{t-1})\|^2 - \eta_\theta\langle \widetilde{g} - \nabla\Phi(\theta_{t-1}), \nabla\Phi(\theta_{t-1})\rangle + (\beta_\theta + \beta_{\theta w}\kappa_{\theta w})\eta_\theta^2\|\widetilde{g}\|^2.$$

Taking expectation (conditional on $\theta_{t-1}, w_{t-1}$),

$$\mathbb{E}[\Phi(\theta_t)] \leqslant \Phi(\theta_{t-1}) - \eta_\theta\|\nabla\Phi(\theta_{t-1})\|^2 - \eta_\theta\langle\nabla_\theta F(\theta_{t-1}, w_{t-1}) - \nabla\Phi(\theta_{t-1}), \nabla\Phi(\theta_{t-1})\rangle$$
$$+ (\beta_\theta + \beta_{\theta w}\kappa_{\theta w})\eta_\theta^2 \left[d_\theta\sigma_\theta^2 + \mathbb{E}\|g - \nabla_\theta F(\theta_{t-1}, w_{t-1})\|^2 + \|\nabla_\theta F(\theta_{t-1}, w_{t-1})\|^2\right]$$
$$\leqslant \Phi(\theta_{t-1}) - \eta_\theta\|\nabla\Phi(\theta_{t-1})\|^2 - \eta_\theta\langle\nabla_\theta F(\theta_{t-1}, w_{t-1}) - \nabla\Phi(\theta_{t-1}), \nabla\Phi(\theta_{t-1})\rangle$$
$$+ (\beta_\theta + \beta_{\theta w}\kappa_{\theta w})\eta_\theta^2 \left[d_\theta\sigma_\theta^2 + \frac{4L_\theta^2}{m}\mathbb{1}_{\{m<n\}} + \|\nabla_\theta F(\theta_{t-1}, w_{t-1})\|^2\right]$$
$$\leqslant \Phi(\theta_{t-1}) - \eta_\theta\|\nabla\Phi(\theta_{t-1})\|^2 - \eta_\theta\langle\nabla_\theta F(\theta_{t-1}, w_{t-1}) - \nabla\Phi(\theta_{t-1}), \nabla\Phi(\theta_{t-1})\rangle$$
$$+ (\beta_\theta + \beta_{\theta w}\kappa_{\theta w})\eta_\theta^2 \left[d_\theta\sigma_\theta^2 + \frac{4L_\theta^2}{m}\mathbb{1}_{\{m<n\}} + 2\|\nabla_\theta F(\theta_{t-1}, w_{t-1}) - \nabla\Phi(\theta_{t-1})\|^2 + 2\|\nabla\Phi(\theta_{t-1})\|^2\right]$$
$$\leqslant \Phi(\theta_{t-1}) - \eta_\theta\|\nabla\Phi(\theta_{t-1})\|^2 + \frac{\eta_\theta}{2}\left[\|\nabla\Phi(\theta_{t-1}) - \nabla_\theta F(\theta_{t-1}, w_{t-1})\|^2 + \|\nabla\Phi(\theta_{t-1})\|^2\right]$$
$$+ (\beta_\theta + \beta_{\theta w}\kappa_{\theta w})\eta_\theta^2 \left[d_\theta\sigma_\theta^2 + \frac{4L_\theta^2}{m}\mathbb{1}_{\{m<n\}} + 2\|\nabla_\theta F(\theta_{t-1}, w_{t-1}) - \nabla\Phi(\theta_{t-1})\|^2 + 2\|\nabla\Phi(\theta_{t-1})\|^2\right]$$
$$\leqslant \Phi(\theta_{t-1}) - \left(\frac{\eta_\theta}{2} - 2(\beta_\theta + \beta_{\theta w}\kappa_{\theta w})\eta_\theta^2\right)\|\nabla\Phi(\theta_{t-1})\|^2$$
$$+ \left(\frac{\eta_\theta}{2} + 2(\beta_\theta + \beta_{\theta w}\kappa_{\theta w})\eta_\theta^2\right)\|\nabla\Phi(\theta_{t-1}) - \nabla_\theta F(\theta_{t-1}, w_{t-1})\|^2$$
$$+ (\beta_\theta + \beta_{\theta w}\kappa_{\theta w})\eta_\theta^2\left(d_\theta\sigma_\theta^2 + \frac{4L_\theta^2}{m}\mathbb{1}_{\{m<n\}}\right).$$

In the first inequality, we used the fact that the Gaussian noise has mean zero and is independent of $(\theta_{t-1}, w_{t-1}, Z)$, plus the fact that $\mathbb{E}g = \nabla_\theta F(\theta_{t-1}, w_{t-1})$. In the second inequality, we used Theorem E.5 and Lipschitz continuity of $f$. In the third and fourth inequalities, we used Young's inequality and Cauchy-Schwartz. □

For the particular $\eta_\theta$ prescribed in Theorem E.3, we obtain:

**Lemma E.8.** *Grant Theorem E.2. If $\eta_\theta = \frac{1}{16\kappa_w(\beta_\theta + \beta_{\theta w}\kappa_{\theta w})}$, then the iterates of Algorithm 2 satisfy* $(\forall t \geqslant 0)$

$$\mathbb{E}\Phi(\theta_{t+1}) \leqslant \mathbb{E}\left[\Phi(\theta_t) - \frac{3}{8}\eta_\theta\|\Phi(\theta_t)\|^2 + \frac{5}{8}\eta_\theta\left(\beta_{\theta w}^2\|w^*(\theta_t) - w_t\|^2 + d_\theta\sigma_\theta^2 + \frac{4L_\theta^2}{m}\mathbb{1}_{\{m<n\}}\right)\right].$$

*Proof.* By Theorem E.7, we have

$$\mathbb{E}\Phi(\theta_{t+1}) \leqslant \mathbb{E}\Phi(\theta_t) - \left(\frac{\eta_\theta}{2} - 2(\beta_\theta + \beta_{\theta w}\kappa_{\theta w})\eta_\theta^2\right)\mathbb{E}\|\nabla\Phi(\theta_t)\|^2$$
$$+ \left(\frac{\eta_\theta}{2} + 2\eta_\theta^2(\beta_\theta + \beta_{\theta w}\kappa_{\theta w})\mathbb{E}\|\nabla\Phi(\theta_t) - \nabla_\theta F(\theta_t, w_t)\|^2\right) + (\beta_\theta\beta_{\theta w}\kappa_{\theta w})\eta_\theta^2\left(d_\theta\sigma_\theta^2 + \frac{4L_\theta^2}{m}\mathbb{1}_{\{m<n\}}\right)$$
$$\leqslant \mathbb{E}\Phi(\theta_t) - \frac{3}{8}\eta_\theta\mathbb{E}\|\nabla\Phi(\theta_t)\|^2 + \frac{5}{8}\eta_\theta\left[\mathbb{E}\|\nabla\Phi(\theta_t) - \nabla_\theta F(\theta_t, w_t)\|^2 + d_\theta\sigma_\theta^2 + \frac{4L_\theta^2}{m}\mathbb{1}_{\{m<n\}}\right]$$
$$\leqslant \mathbb{E}\Phi(\theta_t) - \frac{3}{8}\eta_\theta\mathbb{E}\|\nabla\Phi(\theta_t)\|^2 + \frac{5}{8}\eta_\theta\left[\beta_{\theta w}^2\mathbb{E}\|w^*(\theta_t) - w_t\|^2 + d_\theta\sigma_\theta^2 + \frac{4L_\theta^2}{m}\mathbb{1}_{\{m<n\}}\right].$$

In the second inequality, we simply used the definition of $\eta_\theta$. In the third inequality, we used the fact that $\nabla\Phi(\theta_t) = \nabla_\theta F(\theta_t, w^*(\theta_t))$ (see Theorem E.4) together with Theorem E.2 (part 3). □

Next, we describe the progress made in the $w_t$ updates:

**Lemma E.9.** *Grant Theorem E.2. If $\eta_w = \frac{1}{\beta_w}$, then the iterates of Algorithm 2 satisfy ($\forall t \geqslant 0$)*

$$\mathbb{E}\|w^*(\theta_{t+1}) - w_{t+1}\|^2 \leqslant \left(1 - \frac{1}{2\kappa_w} + 4\kappa_w\kappa_{\theta w}^2\eta_\theta^2\beta_{\theta w}^2\right)\mathbb{E}\|w^*(\theta_t) - w_t\|^2 + \frac{2}{\beta_w^2}\left(\frac{4L_w^2}{m}\mathbb{1}_{\{m<n\}} + d_w\sigma_w^2\right)$$
$$+ 4\kappa_w\kappa_{\theta w}^2\eta_\theta^2\left(\mathbb{E}\|\nabla\Phi(\theta_t)\|^2 + d_\theta\sigma_\theta^2\right).$$

*Proof.* Fix any $t$ and denote $\delta_t := \mathbb{E}\|w^*(\theta_t) - w_t\|^2 := \mathbb{E}\|w^* - w_t\|^2$. We may assume without loss of generality that $f(\theta, \cdot; z)$ is $\mu$-strongly *convex* and that $w_{t+1} = \Pi_{\mathcal{W}}[w_t - \frac{1}{\beta_w}\left(\frac{1}{m}\sum_{i=1}^m \nabla_w f(\theta_t, w_t; z_i) + v_t\right)] := \Pi_{\mathcal{W}}[w_t - \frac{1}{\beta_w}(\nabla h(w_t) + v_t)] := \Pi_{\mathcal{W}}[w_t - \frac{1}{\beta_w}\nabla\tilde{h}(w_t)]$. Now,

$$\mathbb{E}\|w_{t+1} - w^*\|^2 = \mathbb{E}\left\|\Pi_{\mathcal{W}}[w_t - \frac{1}{\beta_w}\nabla\tilde{h}(w_t)] - w^*\right\|^2 \leqslant \mathbb{E}\left\|w_t - \frac{1}{\beta_w}\nabla\tilde{h}(w_t) - w^*\right\|^2$$

$$= \mathbb{E}\|w_t - w^*\|^2 + \frac{1}{\beta_w^2}\left[\mathbb{E}\|\nabla h(w_t)\|^2 + d_w\sigma_w^2\right] - \frac{2}{\beta_w}\mathbb{E}\left\langle w_t - w^*, \nabla\tilde{h}(w_t)\right\rangle$$

$$\leqslant \mathbb{E}\|w_t - w^*\|^2 + \frac{1}{\beta_w^2}\left[\mathbb{E}\|\nabla h(w_t)\|^2 + d_w\sigma_w^2\right] - \frac{2}{\beta_w}\mathbb{E}\left[F(\theta_t, w_t) - F(\theta_t, w^*) + \frac{\mu}{2}\|w_t - w^*\|^2\right]$$

$$\leqslant \delta_t\left(1 - \frac{\mu}{\beta_w}\right) - \frac{2}{\beta_w}\mathbb{E}\left[F(\theta_t, w_t) - F(\theta_t, w^*)\right] + \frac{\mathbb{E}\|\nabla h(w_t)\|^2}{\beta_w^2} + \frac{d_w\sigma_w^2}{\beta_w^2}.$$

Further,

$$\mathbb{E}\|\nabla h(w_t)\|^2 = \mathbb{E}\left[\|\nabla h(w_t) - \nabla_w F(\theta_t, w_t)\|^2 + \|\nabla_w F(\theta_t, w_t)\|^2\right]$$

$$\leqslant \frac{4L_w^2}{m}\mathbb{1}_{\{m<n\}} + \mathbb{E}\|\nabla_w F(\theta_t, w_t)\|^2$$

$$\leqslant \frac{4L_w^2}{m}\mathbb{1}_{\{m<n\}} + 2\beta_w[F(\theta_t, w_t) - F(\theta_t, w^*(\theta_t))],$$

using independence and Theorem E.5 plus Lipschitz continuity of $f$ in the first inequality and Theorem E.6 (plus Theorem E.2 part 5) in the second inequality. This implies

$$\mathbb{E}\|w_{t+1} - w^*\|^2 \leqslant \delta_t\left(1 - \frac{1}{\kappa_w}\right) + \frac{1}{\beta_w^2}\left[d_w\sigma_w^2 + \frac{4L_w^2}{m}\mathbb{1}_{\{m<n\}}\right]. \tag{8}$$

Therefore,

$$\delta_{t+1} = \mathbb{E}\|w_{t+1} - w^*(\theta_t) + w^*(\theta_t) - w^*(\theta_{t+1})\|^2$$

$$\leqslant \left(1 + \frac{1}{2\kappa_w - 1}\right)\mathbb{E}\|w_{t+1} - w^*(\theta_t)\|^2 + 2\kappa_w\mathbb{E}\|w^*(\theta_t) - w^*(\theta_{t+1})\|^2$$

$$\leqslant \left(1 + \frac{1}{2\kappa_w - 1}\right)\left(1 - \frac{1}{\kappa_w}\right)\delta_t + \frac{2}{\beta_w^2}\left[d_w\sigma_w^2 + \frac{4L_w^2}{m}\mathbb{1}_{\{m<n\}}\right] + 2\kappa_w\mathbb{E}\|w^*(\theta_t) - w^*(\theta_{t+1})\|^2$$

$$\leqslant \left(1 + \frac{1}{2\kappa_w - 1}\right)\left(1 - \frac{1}{\kappa_w}\right)\delta_t + \frac{2}{\beta_w^2}\left[d_w\sigma_w^2 + \frac{4L_w^2}{m}\mathbb{1}_{\{m<n\}}\right] + 2\kappa_w\kappa_{\theta w}^2\mathbb{E}\|\theta_t - \theta_{t+1}\|^2$$

$$\leqslant \left(1 + \frac{1}{2\kappa_w - 1}\right)\left(1 - \frac{1}{\kappa_w}\right)\delta_t + \frac{2}{\beta_w^2}\left[d_w\sigma_w^2 + \frac{4L_w^2}{m}\mathbb{1}_{\{m<n\}}\right]$$
$$+ 4\kappa_w\kappa_{\theta w}^2\eta_\theta^2\left[\mathbb{E}\|\nabla_\theta F(\theta_t, w_t) - \nabla\Phi(\theta_t)\|^2 + \|\nabla\Phi(\theta_t)\|^2 + d_\theta\sigma_t^2\right]$$

$$= \left(1 + \frac{1}{2\kappa_w - 1}\right)\left(1 - \frac{1}{\kappa_w}\right)\delta_t + \frac{2}{\beta_w^2}\left[d_w\sigma_w^2 + \frac{4L_w^2}{m}\mathbb{1}_{\{m<n\}}\right]$$
$$+ 4\kappa_w\kappa_{\theta w}^2\eta_\theta^2\left[\mathbb{E}\|\nabla_\theta F(\theta_t, w_t) - \nabla_\theta F(\theta_t, w^*(\theta_t))\|^2 + \|\nabla\Phi(\theta_t)\|^2 + d_\theta\sigma_t^2\right]$$

$$\leqslant \left(1 + \frac{1}{2\kappa_w - 1}\right)\left(1 - \frac{1}{\kappa_w}\right)\delta_t + \frac{2}{\beta_w^2}\left[d_w\sigma_w^2 + \frac{4L_w^2}{m}\mathbb{1}_{\{m<n\}}\right]$$
$$+ 4\kappa_w\kappa_{\theta w}^2\eta_\theta^2\left[\beta_{\theta w}^2\mathbb{E}\|w_t - w^*(\theta_t)\|^2 + \|\nabla\Phi(\theta_t)\|^2 + d_\theta\sigma_t^2\right],$$

by Young's inequality, (8), and Theorem E.4. Since $\left(1 + \frac{1}{2\kappa_w - 1}\right)\left(1 - \frac{1}{\kappa_w}\right) \leqslant 1 - \frac{1}{2\kappa_w}$, we obtain

$$\delta_{t+1} \leqslant \left(1 - \frac{1}{2\kappa_w} + 4\kappa_w\kappa_{\theta w}^2\eta_\theta^2\beta_{\theta w}^2\right)\delta_t + \frac{2}{\beta_w^2}\left[d_w\sigma_w^2 + \frac{4L_w^2}{m}\mathbb{1}_{\{m<n\}}\right] + 4\kappa_w\kappa_{\theta w}^2\eta_\theta^2\left[\|\nabla\Phi(\theta_t)\|^2 + d_\theta\sigma_t^2\right],$$

as desired. $\qquad\square$

We are now prepared to prove Theorem E.3.

*Proof of Theorem E.3.* **Privacy:** This is an easy consequence of Theorem 1 in Abadi et al. (2016) (with precise constants obtained from the proof therein, as in Bassily et al. (2019)) applied to both the min (descent in $\theta$) and max (ascent in $w$) updates. Unlike Abadi et al. (2016), we don't clip the gradients here before adding noise, but the Lipschitz continuity assumptions (Theorem E.2 parts 1 and 2) imply that the $\ell_2$-sensitivity of the gradient updates in lines 4 and 5 of Algorithm 2 are nevertheless bounded by $2L_\theta/m$ and $2L_w/m$, respectively. Thus, Theorem 1 in Abadi et al. (2016) still applies.

**Convergence:** Denote $\zeta := 1 - \frac{1}{2\kappa_w} + 4\kappa_w\kappa_{\theta w}^2\eta_\theta^2\beta_{\theta w}^2$, $\delta_t = \mathbb{E}\|w^*(\theta_t) - w_t\|^2$, and

$$C_t := \frac{2}{\beta_w^2}\left(\frac{4L_w^2}{m}\mathbb{1}_{\{m<n\}} + d_w\sigma_w^2\right) + 4\kappa_w\kappa_{\theta w}^2\eta_\theta^2\left(\mathbb{E}\|\nabla\Phi(\theta_t)\|^2 + d_\theta\sigma_\theta^2\right),$$

so that Theorem E.9 reads as

$$\delta_t \leqslant \zeta\delta_{t-1} + C_{t-1} \tag{9}$$

for all $t \in [T]$. Applying (9) recursively, we have

$$\delta_t \leqslant \zeta^t\delta_0 + \sum_{j=0}^{t-1}C_{t-j-1}\zeta^j$$

$$\leqslant \zeta^t D^2 + 4\kappa_w\kappa_{\theta w}^2\eta_\theta^2\sum_{j=0}^{t-1}\zeta^{t-1-j}\mathbb{E}\|\nabla\Phi(\theta_j)\|^2$$

$$+ \left(\sum_{j=0}^{t-1}\zeta^{t-1-j}\right)\left[\frac{2}{\beta_w^2}\left(\frac{4L_w^2}{m}\mathbb{1}_{\{m<n\}} + d_w\sigma_w^2\right) + 4\kappa_w\kappa_{\theta w}^2\eta_\theta^2 d_\theta\sigma_\theta^2\right].$$

Combining this inequality with Theorem E.8, we get

$$\mathbb{E}\Phi(\theta_t) \leqslant \mathbb{E}\left[\Phi(\theta_{t-1}) - \frac{3}{8}\eta_\theta\|\nabla\Phi(\theta_{t-1})\|^2\right] + \frac{5}{8}\eta_\theta\left(d_\theta\sigma_\theta^2 + \frac{4L_\theta^2}{m}\mathbb{1}_{\{m<n\}}\right)$$

$$+ \frac{5}{8}\eta_\theta\beta_{\theta w}^2\left\{\zeta^t D^2 + 4\kappa_w\kappa_{\theta w}^2\eta_\theta^2\sum_{j=0}^{t-1}\zeta^{t-1-j}\mathbb{E}\|\nabla\Phi(\theta_j)\|^2\right.$$

$$\left.+ \left(\sum_{j=0}^{t-1}\zeta^{t-1-j}\right)\left[\frac{2}{\beta_w^2}\left(\frac{4L_w^2}{m}\mathbb{1}_{\{m<n\}} + d_w\sigma_w^2\right) + 4\kappa_w\kappa_{\theta w}^2\eta_\theta^2 d_\theta\sigma_\theta^2\right]\right\}.$$

Summing over all $t \in [T]$ and re-arranging terms yields

$$\mathbb{E}\Phi(\theta_T) \leqslant \Phi(\theta_0) - \frac{3}{8}\eta_\theta\sum_{t=1}^{T}\mathbb{E}\|\nabla\Phi(\theta_{t-1})\|^2 + \frac{5}{8}\eta_\theta T\left(d_\theta\sigma_t^2 + \frac{4L_\theta^2}{m}\mathbb{1}_{\{m<n\}}\right) + \frac{5}{8}\eta_\theta\beta_{\theta w}^2\left(\sum_{t=1}^{T}\zeta^t\right)D^2$$

$$+ 4\eta_\theta^3\beta_{\theta w}^2\kappa_w\kappa_{\theta w}^2\sum_{t=1}^{T}\sum_{j=0}^{t-1}\zeta^{t-1-j}\mathbb{E}\|\nabla\Phi(\theta_j)\|^2$$

$$+ \frac{5}{8}\left(\sum_{t=1}^{T}\sum_{j=0}^{t-1}\zeta^{t-1-j}\right)\eta_\theta\beta_{\theta w}^2\left[\frac{2}{\beta_w^2}\left(\frac{4L_w^2}{m}\mathbb{1}_{\{m<n\}} + d_w\sigma_w^2\right) + 4\kappa_w\kappa_{\theta w}^2\eta_\theta^2 d_\theta\sigma_\theta^2\right].$$

Now, $\zeta \leqslant 1 - \frac{1}{4\kappa_w}$, which implies that

$$\sum_{t=1}^{T} \zeta^t \leqslant 4\kappa_w \quad \text{and}$$

$$\sum_{t=1}^{T} \sum_{j=0}^{t-1} \zeta^{t-1-j} \leqslant 4\kappa_w T.$$

Hence

$$\frac{1}{T} \sum_{t=1}^{T} \mathbb{E}\|\nabla\Phi(\theta_t)\|^2 \leqslant \frac{3[\Phi(\theta_0) - \mathbb{E}\Phi(\theta_T)]}{\eta_\theta T} + \frac{5}{3}\left(d_\theta\sigma_\theta^2 + \frac{4L_\theta^2}{m}\mathbb{1}_{\{m<n\}}\right) + \frac{7\beta_{\theta w}^2 D^2 \kappa_w}{T}$$
$$+ \frac{48\eta_\theta^2 \beta_{\theta w}^2 \kappa_w^2 \kappa_{\theta w}^2}{T}\left(\sum_{t=1}^{T}\mathbb{E}\|\nabla\Phi(\theta_t)\|^2\right)$$
$$+ 8\kappa_w\beta_{\theta w}^2 \frac{2}{\beta_w^2}\left(\frac{4L_w^2}{m}\mathbb{1}_{\{m<n\}} + d_w\sigma_w^2\right) + 32\beta_{\theta w}^2 \kappa_w^2 \kappa_{\theta w}^2 \eta_\theta^2 d_\theta\sigma_\theta^2.$$

Since $\eta_\theta^2 \beta_{\theta w}^2 \kappa_w^2 \kappa_{\theta w}^2 \leqslant \frac{1}{256}$, we obtain

$$\mathbb{E}\|\nabla\Phi(\hat{\theta}_T)\|^2 \lesssim \frac{\Delta_\Phi \kappa_w}{T}(\beta_\theta + \beta_{\theta w}\kappa_{\theta w}) + \frac{d_\theta L_\theta^2 T \ln(1/\delta)}{\epsilon^2 n^2} + \frac{1}{m}\mathbb{1}_{\{m<n\}}\left(L_\theta^2 + \frac{\kappa_w\beta_{\theta w}^2 L_w^2}{\beta_w^2}\right) + \frac{\kappa_w\beta_{\theta w}^2 L_w^2 d_w T \ln(1/\delta)}{\beta_w^2 \epsilon^2 n^2}$$
$$+ \frac{\beta_{\theta w}^2 D^2 \kappa_w}{T}.$$

Our choice of $T$ then implies

$$\mathbb{E}\|\nabla\Phi(\hat{\theta}_T)\|^2 \lesssim \sqrt{\Delta_\Phi(\beta_\theta + \beta_{\theta w}\kappa_{\theta w})\kappa_w + \kappa_w\beta_{\theta w}^2 D^2}\left[\frac{L_\theta\sqrt{d_\theta\ln(1/\delta)}}{\epsilon n} + \left(\frac{\beta_{\theta w}\sqrt{\kappa_w}}{\beta_w}\right)\frac{L_w\sqrt{d_w\ln(1/\delta)}}{\epsilon n}\right]$$
$$+ \frac{\mathbb{1}_{\{m<n\}}}{m}\left(L_\theta^2 + \frac{\kappa_w\beta_{\theta w}^2 L_w^2}{\beta_w^2}\right).$$

Finally, our choice of sufficiently large $m$ yields the last claim in Theorem E.3. $\qquad\square$

### E.2    Proof of Theorem 3.4

Theorem 3.4 is an easy consequence of Theorem E.1, which we proved in the above subsection:

**Theorem E.10** (Re-statement of Theorem 3.4). *Assume the loss function $\ell(\cdot, x, y)$ and $\mathcal{F}(x, \cdot)$ are Lipschitz continuous with Lipschitz gradient for all $(x, y)$, and $\widehat{P}_S(r) \geqslant \rho > 0 \; \forall \; r \in [k]$. In Algorithm 1, choose $\mathcal{W}$ to be a sufficiently large ball that contains $W^*(\theta) := \operatorname{argmax}_W \widehat{F}(\theta, W)$ for every $\theta$ in some neighborhood of $\theta^* \in \operatorname{argmin}_\theta \max_W \widehat{F}(\theta, W)$. Then there exist algorithmic parameters such that the $(\epsilon, \delta)$-DP Algorithm 1 returns $\hat{\theta}_T$ with*

$$\mathbb{E}\|\nabla FERMI(\hat{\theta}_T)\|^2 = \mathcal{O}\left(\frac{\sqrt{\max(d_\theta, kl)\ln(1/\delta)}}{\epsilon n}\right),$$

*treating $D = diameter(\mathcal{W})$, $\lambda$, $\rho$, $l$, and the Lipschitz and smoothness parameters of $\ell$ and $\mathcal{F}$ as constants.*

*Proof.* By Theorem E.1, it suffices to show that $f(\theta, W; z_i) := \ell(\theta, x_i, y_i) + \lambda\widehat{\psi}_i(\theta, W)$ is Lipschitz continuous with Lipschitz gradient in both the $\theta$ and $W$ variables for any $z_i = (x_i, y_i, s_i)$, and that $f(\theta, \cdot; z_i)$ is strongly concave. We assumed $\ell(\cdot, x_i, y_i)$ is Lipschitz continuous with Lipschitz gradient. Further, the work of Lowy et al. (2022a) showed that $f(\theta, \cdot; z_i)$ is strongly concave. Thus, it suffices to show that $\widehat{\psi}_i(\theta, W)$ is Lipschitz continuous with Lipschitz gradient. This clearly holds by Theorem D.1, since $\mathcal{F}(x, \cdot)$ is Lipschitz continuous with Lipschitz gradient and $W \in \mathcal{W}$ is bounded. $\qquad\square$

# F  NUMERICAL EXPERIMENTS: ADDITIONAL DETAILS AND RESULTS

## F.1  MEASURING DEMOGRAPHIC PARITY AND EQUALIZED ODDS VIOLATION

We used the expressions given in (10) and (11) to measure the demographic parity violation and the equalized odds violation respectively. We denote $\mathcal{Y}$ to be the set of all possible output classes and $\mathcal{S}$ to be the classes of the sensitive attribute. $P[E]$ denotes the empirical probability of the occurrence of an event E.

$$\max_{y' \in \mathcal{Y}, s_1, s_2 \in \mathcal{S}} \left| P[\widehat{y} = y' | s = s_1] - P[\widehat{y} = y' | s = s_2] \right| \tag{10}$$

$$\max_{y' \in \mathcal{Y}, s_1, s_2 \in \mathcal{S}} \max( \left| P[\widehat{y} = y' | s = s_1, y = y'] - P[\widehat{y} = y' | s = s_2, y = y'] \right|,$$
$$\left| P[\widehat{y} = y' | s = s_1, y \neq y'] - P[\widehat{y} = y' | s = s_2, y \neq y'] \right|) \tag{11}$$

## F.2  TABULAR DATASETS

### F.2.1  MODEL DESCRIPTION AND EXPERIMENTAL DETAILS

**Demographic Parity:** We split each dataset in a 3:1 train:test ratio. We preprocess the data similar to Hardt et al. (2016a) and use a simple logistic regression model with a sigmoid output $O = \sigma(Wx + b)$ which we treat as conditional probabilities $p(\widehat{y} = i | x)$. The predicted variables and sensitive attributes are both binary in this case across all the datasets. We analyze fairness-accuracy trade-offs with four different values of $\epsilon \in \{0.5, 1, 3, 9\}$ for each dataset. We compare against state-of-the-art algorithms proposed in Tran et al. (2021a) and (the demographic parity objective of) Tran et al. (2021b). The tradeoff curves for DP-FERMI were generated by sweeping across different values for $\lambda \in [0, 2.5]$. The learning rates for the descent and ascent, $\eta_\theta$ and $\eta_w$, remained constant during the optimization process and were chosen from $[0.005, 0.01]$. Batch size was 1024. We tuned the $\ell_2$ diameter of the projection set $\mathcal{W}$ and $\theta$-gradient clipping threshold in $[1, 5]$ in order to generate stable results with high privacy (i.e. low $\epsilon$). Each model was trained for 200 epochs. The results displayed are averages over 15 trials (random seeds) for each value of $\epsilon$.

**Equalized Odds:** We replicated the experimental setup described above, but we took $\ell_2$ diameter of $\mathcal{W}$ and the value of gradient clipping for $\theta$ to be in $[1, 2]$. Also, we only tested three values of $\epsilon \in \{0.5, 1, 3\}$.

### F.2.2  DESCRIPTION OF DATASETS

**Adult Income Dataset:** This dataset contains the census information about the individuals. The classification task is to predict whether the person earns more than 50k every year or not. We followed a preprocessing approach similar to Lowy et al. (2022a). After preprocessing, there were a total of 102 input features from this dataset. The sensitive attribute for this work in this dataset was taken to be gender. This dataset consists of around 48,000 entries spanning across two CSV files, which we combine and then we take the train-test split of 3:1.

**Retired Adult Income Dataset:** The Retired Adult Income Dataset proposed by Ding et al. (2021a) is essentially a superset of the Adult Income Dataset which attempts to counter some caveats of the Adult dataset. The input and the output attributes for this dataset is the same as that of the Adult Dataset and the sensitive attribute considered in this work is the same as that of the Adult. This dataset contains around 45,000 entries.

**Parkinsons Dataset:** In the Parkinsons dataset, we use the part of the dataset which had the UPRDS scores along with some of the features of the recordings obtained from individuals affected and not affected with the Parkinsons disease. The classification task was to predict from the features whether the UPDRS score was greater than the median score or not. After preprocessing, there were a total of 19 input features from this dataset and the sensitive attribute for this dataset was taken to be gender. This dataset contains around 5800 entries in total. We took a train-test split of 3:1.

**Credit Card Dataset:** This dataset contains the financial data of users in a bank in Taiwan consisting of their gender, education level, age, marital status, previous bills, and payments. The assigned

classification task is to predict whether the person defaults their credit card bills or not, essentially making the task if the clients were credible or not. We followed a preprocessing approach similar to Lowy et al. (2022a). After preprocessing, there were a total of 85 input features from this dataset. The sensitive attribute for this dataset was taken to be gender. This dataset consists of around 30,000 entries from which we take the train-test split of 3:1.

**UTK-Face Dataset:** This dataset is a large scale image dataset containing with an age span from 0 to 116. The dataset consists of over 20,000 face images with details of age, gender, and ethnicity and covers large variation in pose, facial expression, illumination, occlusion, resolution. We consider the age classification task with 9 age groups similar to the experimental setup in Tran et al. (2022). We consider the sensitive attribute to be the ethnicity which consists of 5 different classes.

### F.2.3 DEMOGRAPHIC PARITY

**Retired Adult Results:** See Fig. 6 for our results on Retired Adult Dataset. The results are qualitatively similar to the reusults reported in the main body: our algorithm (DP-FERMI) achieves the most favorable fairness-accuracy tradeoffs across all privacy levels.

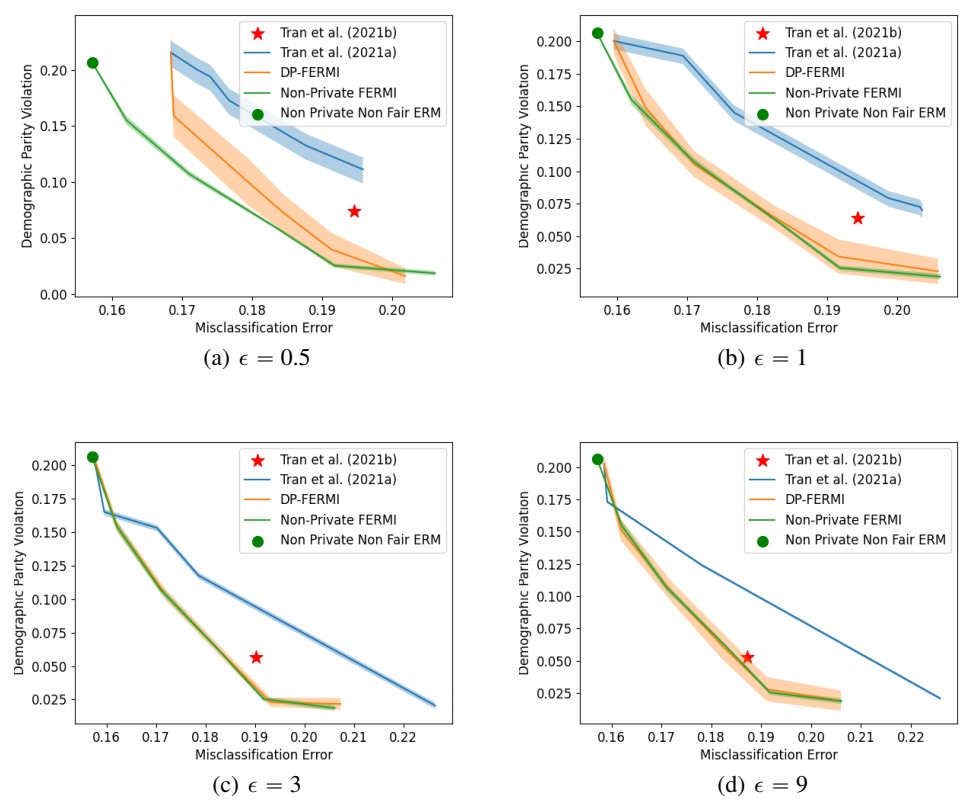

Figure 6: Private, fair logistic regression on the Retired Adult Dataset

**Credit Card Results:** See Fig. 7 for our results on Credit Card Dataset. DP-FERMI offers superior fairness-accuracy-privacy profile compared to all applicable baselines.

**Additional Results for Parkinsons Dataset:** More results for Parkinsons are shown in Fig. 8. DP-FERMI offers the best performance.

### F.2.4 EQUALIZED ODDS

**Equalized Odds Variation of DP-FERMI Algorithm:** The (FERMI obj.) minimizes the Exponential Renyi Mutual Information (ERMI) between the output and the sensitive attributes which

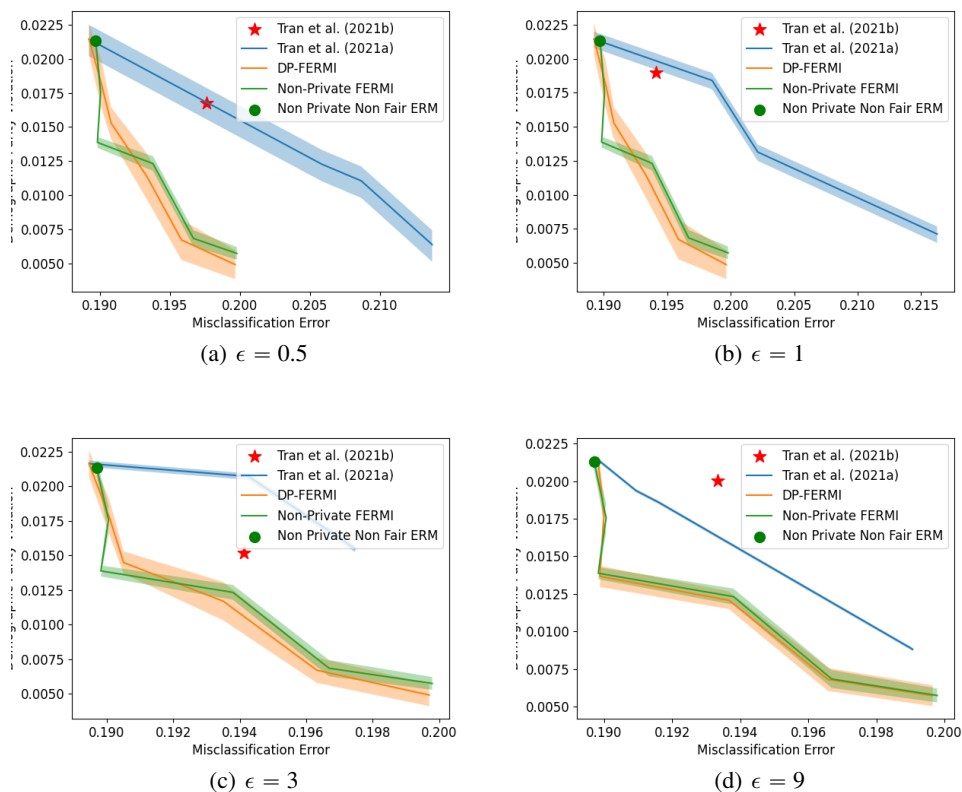

Figure 7: Private, fair (demographic parity) logistic regression on the Credit Card Dataset

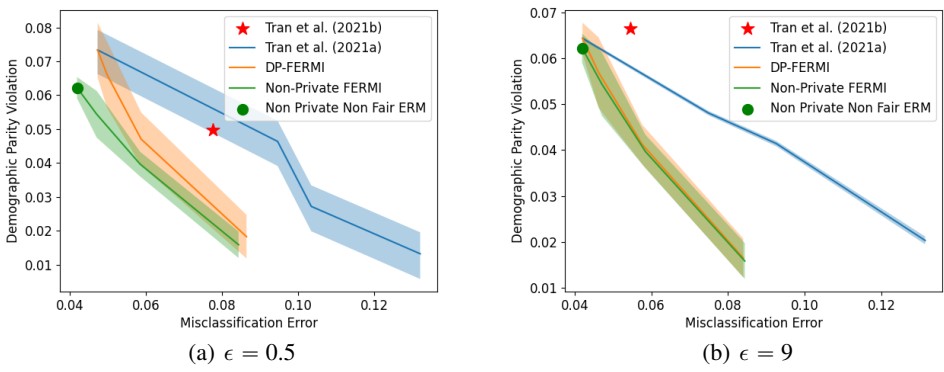

Figure 8: Private, Fair (Demogrpahic Parity) Logistic regression on Parkinsons Dataset

essentially leads to a reduced demographic parity violation. The equalized-odds condition is more constrained and enforces the demographic parity condition for data grouped according to labels. For the equalized-odds, the ERMI between the predicted and the sensitive attributes is minimized conditional to each of the label present in the output variable of the dataset. So, the FERMI regularizer is split into as many parts as the number of labels in the output. This enforces each part of the FERMI regularizer to minimize the ERMI while an output label is given/constant. Each part has its own unique W that is maximized in order to create a stochastic estimator for the ERMI with respect to each of the output labels.

**Adult Results:** Results for the equalized odds version of DP-FERMI on Adult dataset are shown in Fig. 9. Our approach outperforms the previous state-of-the-art methods.

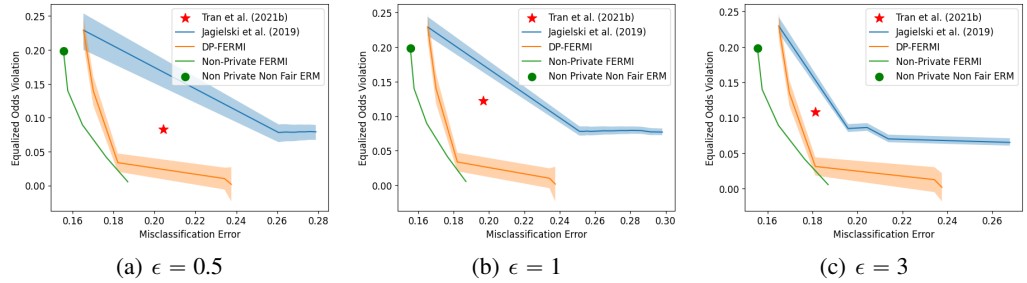

(a) $\epsilon = 0.5$      (b) $\epsilon = 1$      (c) $\epsilon = 3$

Figure 9: Results obtained for applying DP-FERMI with equalized odds violation on logistic regression on the Adult Dataset

**Retired Adult Results:** (Initial) Results for the equalized odds version of DP-FERMI on the retired-adult dataset are shown in Fig. 10. Our approach outperforms Tran et al. (2021b) and we are currently tuning our non-private and/or non-fair versions of our models along with Jagielski et al. (2019).

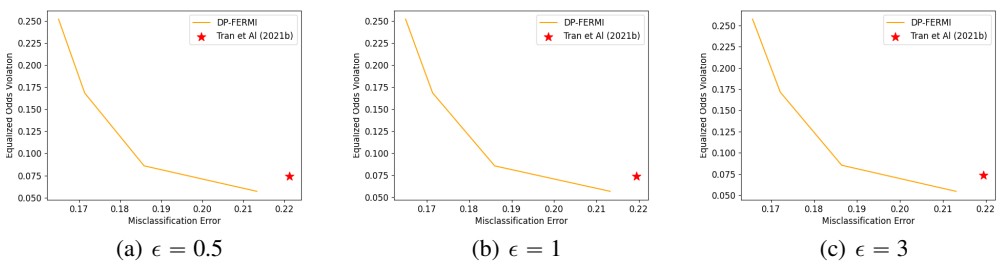

(a) $\epsilon = 0.5$      (b) $\epsilon = 1$      (c) $\epsilon = 3$

Figure 10: Results obtained for applying DP-FERMI with equalized odds violation on logistic regression on the Retired Adult Dataset

### F.3 IMAGE DATASET (UTK-FACE)

We split the dataset in a 3:1 train:test ratio. Batch size was 64. 128 x 128 normalized images were used as input for our model. We tuned the $\ell_2$ diameter of $\mathcal{W}$ and the value of gradient clipping for $\theta$ to be in $[1, 2]$ and learning rates for the descent and ascent, $\eta_\theta$ and $\eta_w$, remained constant during the optimization process and were chosen as 0.001 and 0.005 respectively. We analyze the fairness-accuracy trade-offs with five different values of $\epsilon \in \{10, 25, 50, 100, 200\}$. The results displayed were averaged over observations obtained from 5 different randomly chosen seeds on each configuration of $\epsilon$ and a dataset. Each model was trained for 150 epochs. The tradeoff curves for this set of experiments were obtained by sweeping across different values for $\lambda \in [0, 500]$.

# G  SOCIETAL IMPACTS

In this paper, we considered the socially consequential problem of *privately* learning *fair* models from sensitive data. Motivated by the lack of *scalable* private fair learning methods in the literature, e developed the first differentially private (DP) fair learning algorithm that is guaranteed to converge with small batches (*stochastic optimization*). We hope that our method will be used to help companies, governments, and other organizations to responsibly use sensitive, private data. Specifically, we hope that our DP-FERMI algorithm will be useful in reducing discrimination in algorithmic decision-making while simultaneously preventing leakage of sensitive user data. The stochastic nature of our algorithm might be especially appealing to companies that are using very large models and datasets. On the other hand, there are also some important limitations of our method that need to be considered before deployment.

One caveat of our work is that we have assumed that the given data set contains fair and accurate labels. For example, if gender is the sensitive attribute and "likelihood of repaying a loan" is the target, then we assume that the training data *accurately* describes everyone's financial history without discrimination. If training data is biased against a certain demographic group, then it is possible that our algorithm could *amplify* (rather than mitigate) unfairness. See e.g. Kilbertus et al. (2020); Bechavod et al. (2019) for further discussion.

Another important practical consideration is how to weigh/value the different desiderata (privacy, fairness, and accuracy) when deploying our method. As shown in prior works (e.g., Cummings et al. (2019)) and re-enforced in the present work, there are fundamental tradeoffs between fairness, accuracy, and privacy: improvements in one generally come at a cost to the other two. Determining the relative importance of each of these three desiderata is a critical question that lacks a clear or general answer. Depending on the application, one might be seriously concerned with either discrimination or privacy attacks, and should calibrate $\epsilon$ and $\lambda$ accordingly. Or, perhaps very high accuracy is necessary for a particular task, with privacy and/or fairness as an afterthought. In such a case, one might want to start with very large $\epsilon$ and small $\lambda$ to ensure high accuracy, and then gradually shrink $\epsilon$ and/or increase $\lambda$ to improve privacy/fairness until training accuracy dips below a critical threshold. A thorough and rigorous exploration of these issues could be an interesting direction for future work.

