# OpenReview forum: "Stochastic Differentially Private and Fair Learning"
_ICLR.cc/2023/Conference — ICLR 2023 poster_

### Official Review · Reviewer_8e2X · 2022-10-14

**Confidence:** 4
**Correctness:** 3
**Technical Novelty And Significance:** 2
**Empirical Novelty And Significance:** 3
**Recommendation:** 3

**Clarity, Quality, Novelty And Reproducibility:**

Overall, the paper is easy to follow. Some results in this paper rely on prior work [Lowy et al., 2021]. It is also unclear if the convergence guarantee derived in this paper is useful in practice.

**Details Of Ethics Concerns:**

Some statements in this paper may be misleading.
Quote from the paper: "if sex is the sensitive attribute and a data set contains all men, then any classifier is trivially fair with respect to sex, so fairness with respect to sex is meaningless for that data set."
This statement makes me uncomfortable. A data set contains all men DOES NOT mean any classifier trained from this data set will be fair. In fact, lacking sample diversity could be one big issue in practice.

**Strength And Weaknesses:**

The problem studied in this paper (i.e., fair learning algorithm with DP guarantees) is important and significant. Overall, this paper is easy to follow. The new algorithms introduced in this paper look interesting.


My main concerns are listed below:

**Privacy mechanism**: it is unclear to me why the DP is defined only in terms of the sensitive attribute S. This may be vulnerable: when S is correlated with the rest of the features, an adversary can potentially reconstruct it from the rest of the features.

**Convergence guarantee**: The convergence guarantee (e.g., Theorem 3.2) is different from the typical convergence results. First, it only guarantees the existence of T (number of iterations), eta_theta, and eta_w (step size) such that the results hold. Hence, it is very likely that the (T, eta_theta, eta_w) a data scientist chooses does not have any convergence guarantee. Second, it is unclear to me why the result is only derived for the gradient of Phi. This is not even the gradient used for updating the parameters. Ideally, the convergence results should tell us the convergence of theta and w. Finally, the authors did not provide any convergence rate so it is unclear how the algorithm converges w.r.t. the number of iterations.

**Estimation error** it seems that the algorithm introduced by the author depends on estimating the joint distribution of \hat{Y} and S. It is unclear to me how the estimation error influences the performance of the algorithm. In particular, since fairness is defined through chi-square divergence and the accuracy of estimating chi-square divergence relies on the support set, I am a bit worried about how the estimation error propagates to the learning algorithm. This problem is important because this paper considers the setting of multi-class, multi-sensitive attributes (I assume this is one main contribution of this paper). Hence, the support of \hat{Y} and S can be very large.

**Misleading statement** the authors wrote “if sex is the sensitive attribute and a data set contains all men, then any classifier is trivially fair with respect to sex, so fairness with respect to sex is meaningless for that data set.” I find this sentence very misleading! The authors should revise this statement.

**Novelty** It seems that many results in this paper rely on prior work [Lowy et al., 2021].

Other comments:

The authors may consider discussing what group fairness measures can be covered in this paper. For example, can the learning algorithm cover false discovery rate and calibration error?

Minor comment: D_R(\hat{Y}, S) is chi-square information. Calling it Renyi mutual information may not be accurate


**Summary Of The Paper:**

This paper introduces a fair learning algorithm with DP guarantees. The algorithm owns a provable “convergence” guarantee and can deal with multi-class, multi-sensitive attributes classification tasks.

**Summary Of The Review:**

Interesting paper but many concerns need to be addressed

---

> ### Author Response · Authors · 2022-11-19
> **Response to Reviewer 8e2X, Part 1**
>
> Thank you for your thoughtful review and constructive feedback! We respond to each of your points below. Due to character limits, we break our response into two parts.
>
> >it is unclear to me why the DP is defined only in terms of the sensitive attribute S. This may be vulnerable: when S is correlated with the rest of the features, an adversary can potentially reconstruct it from the rest of the features.
>
> You have raised a great point! We followed the standard setup in the DP fair learning literature going back to Jagielski et al. (2019) which only seeks to privatize the sensitive attributes. The motivation for this setup is that we want to train a fair model, but might not be able to use the sensitive attributes (which are needed to train a fair model) due to privacy concerns/laws (e.g. the EU’s GDPR). The solution proposed by Jagielski et al. (2019) is to ensure the differential privacy of the sensitive data, so that the company can safely use these sensitive attributes to train a fair model, while satisfying legal and ethical constraints on user privacy.
>
> That being said, you make a good point: there are certainly cases where we might want/need to ensure the privacy of features other than just the sensitive attributes. Fortunately, **our algorithm and analysis easily extends to the case where all the features are private**. To ensure differential privacy for all the features, the only change is that the noise variance $\sigma^2$ in Algorithm 2 would need to increase because the *sensitivity* estimates for the gradient updates would increase. Our stochastic Algorithm 2 still converges at the same asymptotic rate given in Corollary 3.1 when $d_{\theta} \gg kl$, as is typical in practical large-scale fairness problems; the rate only gets slower by constant factors and factors depending on $k$ and $l$. We are currently working on typing this up and including it in the revision.
>
> >Convergence guarantee: The convergence guarantee (e.g., Theorem 3.2) is different from the typical convergence results. First, it only guarantees the existence of T (number of iterations), eta_theta, and eta_w (step size) such that the results hold. Hence, it is very likely that the (T, eta_theta, eta_w) a data scientist chooses does not have any convergence guarantee.
>
> This is standard in theoretical analysis for DP optimization. For example, even for the simpler nonconvex *minimization* problem,  all existing DP algorithms do not provide any non-trivial guarantee on the stationarity gap as $T \to \infty$ (see e.g. Arora et al., 2022; Tran and Cutkosky, 2022; and the references therein). This is because the privacy noise variance increases linearly with $T$, making the stationarity gap bound a non-monotonic function of $T$. Please let us know if our explanation answers your question.
>
> >Convergence guarantee…it is unclear to me why the result is only derived for the gradient of Phi. This is not even the gradient used for updating the parameters. Ideally, the convergence results should tell us the convergence of theta and w.
>
> The norm of the gradient of $\Phi$ is a standard measure of convergence in nonconvex (min-max) optimization: see for example, Lin et al., 2020; Ostrovskii et al., 2021 and the discussion therein. **In our application, $\nabla \Phi (\theta) = \nabla FERMI(\theta)$, where FERMI is the fair ERM objective function that we are aiming to minimize. Thus, $\|\nabla \Phi(\theta_t)\|$ is a natural and meaningful measure of stationarity in our problem**. See Corollary 3.1. Having said that, for certain classes of functions, $\|\nabla \Phi(\theta_t)\|$ can be related to loss function value (e.g.  functions satisfying the Polyak- Lojasiewicz condition (Polyak, 1963; Karimi et al., 2016) or strongly convex functions) . However, **for general non-convex loss functions, efficiently minimizing the function value or the distance to the optimum $\theta^*$ is NP-hard** (Murty, 1985).
>
> >the authors did not provide any convergence rate so it is unclear how the algorithm converges w.r.t. the number of iterations.
>
> In the formal version of Theorem 3.2—**Theorem B.1** in the Appendix–-we provide the number of iterations $T$ needed to achieve the stationarity guarantee. We mention this in the revision, after the statement of Theorem 3.2.  Please let us know if this answers your question.

---

> > ### Author Response · Authors · 2022-11-19
> > **Response to Reviewer 8e2X, Part 2**
> >
> > > Estimation error it seems that the algorithm introduced by the author depends on estimating the joint distribution of \hat{Y} and S. It is unclear to me how the estimation error influences the performance of the algorithm…
> >
> > This is a great question! Let us answer this question by borrowing some results from prior work: Lowy et al. (2021) [Proposition 2] show that **empirical ERMI (Definition 3) is a consistent and asymptotically unbiased estimator of *“population ERMI”***--which is defined as in Definition 3 except that empirical distributions are replaced by their population counterparts. This suggests that with enough i.i.d. samples at our disposal, the solution that our fair ERM algorithm converges to will be a good approximation of the solution of the corresponding fair population risk minimization problem. Moreover, this is supported empirically in our paper: **our experiments show that DP-FERMI algorithm archives low *test error*** under DP and fairness constraints. We added this discussion to the rebuttal revision.
> >
> > >Misleading statement the authors wrote “if sex is the sensitive attribute and a data set contains all men, then any classifier is trivially fair with respect to sex, so fairness with respect to sex is meaningless for that data set.” I find this sentence very misleading! The authors should revise this statement.
> >
> > Thank you for bringing this to our attention. What we meant to say was “...trivially fair with respect to sex *on the training data*...” However, you are right that the resulting classifier may still be unfair with respect to test data. In fact, as you mention, **diverse representation in training data is crucial for training a fair model** (with respect to test data). Thus, the assumption on $\rho$ is practically reasonable. Understanding what performance is (im-)possible for DP fair learning in the absence of sample diversity is an interesting and important direction for future work. We addressed your concern in our revision. We again thank you for bringing this to our attention and please let us know if your concern is not addressed.
> >
> > >Novelty: It seems that many results in this paper rely on prior work [Lowy et al., 2021].
> >
> > Like most  DP fair learning works, the main novelty of our paper is in **privatizing** an existing fair learning algorithm. In our case, we privatize the FERMI algorithm of [Lowy et al., 2021] and prove that it converges. Privately solving the nonconvex min-max problem that arises from Theorem 3.1 is nontrivial. Indeed, no convergent DP algorithm for solving nonconvex min-max optimization problems was known to exist prior to our paper. Thus, our Theorem 3.2 is a novel contribution to the DP optimization literature, independently of our application to DP fair learning.
> >
> > >The authors may consider discussing what group fairness measures can be covered in this paper. For example, can the learning algorithm cover false discovery rate and calibration error?
> >
> > We would like to thank the reviewer for pointing this out. The scope of our work covers *independence condition metrics(such as demographic parity or any modified demographic parity metric) and separation condition metrics (such as equalized odds, equality of opportunity, and any other variants)* in the fairness literature. Our regularizer can model independence metrics between two probability distributions, but when we convert the estimator to an unbiased estimator compatible with stochastic optimization, it requires the predicted label terms to stay in the numerator so that the expectation can be expressed in terms of the predicted probability and our loss function can be formulated. Then, any partitions based on the actual label can be accounted for in terms of an external summation to the given expression. We have added the categories of group fairness metrics in our rebuttal revision.
> >
> > Since false discovery rate and calibration errors are specific cases of sufficiency metrics, hence, it seems they cannot be modeled by our framework. Moreover, there has been a work (Pleiss, 2017) that proves that an algorithm satisfying the calibration is no better than randomizing a certain percentage of predictions. However, this does indicate an important question in the future on whether it is (im-)possible to develop provably stochastic frameworks that reduce violations on other sufficiency metrics.
> >
> > > Minor comment: D_R(\hat{Y}, S) is chi-square information. Calling it Renyi mutual information may not be accurate
> >
> > *Exponential* Renyi mutual information (ERMI) is terminology borrowed from [Lowy et al. 2021].  You are right that it is also equal to chi-square information.

---

> > ### Comment · Reviewer_8e2X · 2022-11-22
> > **Thanks for the response & follow-up comments**
> >
> > First, I would like to thank the authors for providing a comprehensive response to my comments! I still have some questions that have not been addressed after the rebuttal:
> >
> > [Privacy mechanism]. I understand that this setup can be traced back to Jagielski et al 2019 and probably existing literature also leverages this privacy notion to mask sensitive attributes. However, it is still unclear to me if this is a reasonable way to protect data privacy. As I wrote in my original review, it is very likely that an adversary can ignore the perturbed sensitive attributes and use the remaining features to predict the sensitive attributes reliably (e.g., [Sweeney 2002]). I am glad to know that the proposed algorithm can extend to protect all features. However, it is unclear to me how easy it is for making this extension (e.g., how the sensitivity changes).
> >
> > [Convergence guarantee] It is still unclear to me if the convergence guarantees derived in this paper are useful or not. As I wrote in my original review, Theorem 3.2 only guarantees the existence of T, eta_theta, and eta_w. In practice, it is very likely that the (T, eta_theta, eta_w) a data scientist chooses does not have any convergence guarantee. Ideally, it would be great to have a convergence bound that holds for any large enough T (e.g., results alike [Raginsky et al 2017]).
> >
> >
> > [Algorithm’s utility] “The final model parameters are chosen by drawing uniformly at random from all intermediate models during training. Won’t that often destroy model utility, if an early iterate is chosen?” This question was originally raised by Reviewer bxYf. In the rebuttal, the authors wrote “For example, we are not aware of any results that prove convergence in expectation for the last iterate of SGD (although almost sure convergence does hold).” However, the proposed algorithm in this paper is more like DP-SGD (or SGLD) rather than SGD. Establishing convergence guarantees for DP-SGD is easier than SGD because the Gaussian noise can potentially help the algorithm escape saddle points. I include one reference below, but I think you can find more references that establish convergence guarantees for DP-SGD when the algorithm outputs the last iterate.
> >
> > Raginsky, M., Rakhlin, A. and Telgarsky, M., 2017, June. Non-convex learning via stochastic gradient langevin dynamics: a nonasymptotic analysis. In Conference on Learning Theory(pp. 1674-1703). PMLR.
> >
> > As a side note, there is a line of work in DP literature studying privacy amplification by iteration. They investigate how outputting the last iterate amplifies privacy guarantees compared with outputting the entire trajectory.
> >
> > [Fairness measures]. In the revised paper, the authors included “In general, our method can minimize all independence metrics, e.g. demographic parity, and all separation metrics, e.g. equalized odds.” I think it is worth making it clear what kind of group fairness metrics cannot be covered by the analysis of this paper. For example, as the authors acknowledged in the rebuttal, false discovery rate and calibration error cannot be covered. In fact, some existing frameworks can incorporate these metrics into their framework:
> >
> > Celis, L.E., Huang, L., Keswani, V. and Vishnoi, N.K., 2019, January. Classification with fairness constraints: A meta-algorithm with provable guarantees. In Proceedings of the conference on fairness, accountability, and transparency (pp. 319-328).
> >
> > Kim, J.S., Chen, J. and Talwalkar, A., 2020, November. Fact: A diagnostic for group fairness trade-offs. In International Conference on Machine Learning (pp. 5264-5274). PMLR.

---

> > > ### Author Response · Authors · 2022-11-23
> > > **Response to your follow-up questions (part 1)**
> > >
> > > Thank you for your feedback and for giving us an opportunity to further clarify your remaining questions! We address each of these points below. Due to character limits, we break our response into two parts.
> > >
> > > >[Privacy mechanism] it is still unclear to me if [differential privacy of sensitive attributes] is a reasonable way to protect data privacy [of sensitive attributes]…it is very likely that an adversary can ignore the perturbed sensitive attributes and use the remaining features to predict the sensitive attributes reliably (e.g., [Sweeney 2002]).
> > >
> > > We assume that the raw data is never disclosed and our algorithm only returns the model parameters. This assumption is reasonable in most applications and is also standard in DP literature.  Thus, the adversary does not have any knowledge of the individuals’ non-sensitive attributes (beyond what can be inferred from public population statistics). Thus, for any such adversary who only observes the output of Algorithm 2, our differential privacy guarantee (Proposition 3.1) means that **the adversary cannot infer “much” more about the sensitive attributes of any individual in the data set than they could have inferred had that individual never contributed their data (Kasiviswanathan and Smith, 2014)**; here, “much” is quantified by $\epsilon$ and $\delta$. Also, see Jagielski et al. (2019) for further discussion.
> > >
> > > That being said, if an adversary had prior knowledge about non-sensitive (or sensitive) features of individuals in the data set, then your concern would be valid.
> > >
> > > Reference:
> > > Shiva P. Kasiviswanathan and Adam Smith. "On the 'semantics' of differential privacy: A bayesian formulation." Journal of Privacy and Confidentiality 6.1 (2014).
> > >
> > > >[Privacy mechanism] I am glad to know that the proposed algorithm can extend to protect all features. However, it is unclear to me how easy it is for making this extension (e.g., how the sensitivity changes).
> > >
> > > The extension to protecting privacy of all features is indeed straightforward. In the notation/setting of Proposition 3.1, we have the following guarantee for Algorithm 2 if all features must remain private:
> > >
> > > **Proposition:** Run Algorithm 2 with privacy noise variances $\Sigma_{\theta}^2 := 4 \sigma_\theta^2 +  \frac{32 D^4 L_{\theta}^2 l^2 T \log(1/\delta)}{\epsilon^2 n^2}$ and $\Sigma_w^2 := 4 \sigma_w^2 + \frac{32 D^4 K^2 T \log(1/\delta)}{\epsilon^2 n^2}$. Then, the full data set (all features) is $(\epsilon, \delta)$-differentially private.
> > >
> > > The proof is very similar to the proof of Proposition 3.1: we use the moments accountant and bound the sensitivity of the gradient estimates. A straightforward computation (using the expressions for the gradients of $\hat{\psi}$ derived in the proof of Proposition 3.1) shows that the new $\theta$-sensitivity (for privacy of all features) is upper bounded by $2 \Delta_{\theta} + \frac{2 D^2 L_{\theta} l}{m}$, where $\Delta_{\theta}$ is the sensitivity estimate derived in the proof of Proposition 3.1. Further, the new $W$-sensitivity (for privacy of all features) is upper bounded by $2 \Delta_{w}  + \frac{2 D^2 K}{m}$.

---

> > > > ### Author Response · Authors · 2022-11-23
> > > > **Response to your follow-up questions (part 2)**
> > > >
> > > > >[Convergence guarantee] It is still unclear to me if the convergence guarantees derived in this paper are useful or not...Theorem 3.2 only guarantees the existence of T, eta_theta, and eta_w. In practice, it is very likely that the (T, eta_theta, eta_w) a data scientist chooses does not have any convergence guarantee. Ideally, it would be great to have a convergence bound that holds for any large enough T (e.g., results alike [Raginsky et al 2017]).
> > > >
> > > > **The data scientist should use the $T$ and $\eta_\theta, \eta_w$ that we prescribe in Theorem B.1 to guarantee convergence**. Although some of the smoothness and Lipschitz parameters might not be known a priori, for fixed loss function it is not hard to get a reasonable upper bound or (with a bit more work) a private estimate of these quantities, with at most constant factor error. The same guaranteed convergence rate then holds up to constant factors.
> > > >
> > > > Addressing privacy concerns properly is at the heart of future ML systems. The data scientist must not choose $T$ arbitrarily as they may violate the privacy of the users, unless $\sigma^2$ increases proportionally. **The reference [Raginsky et al 2017] is not concerned with privacy**, so their noise variance does not increase with $T$. Thus, as is standard in non-private optimization, the utility of their algorithm improves (or at least does not degrade) as $T$ increases. By contrast, **all existing (iterative) DP algorithms for non-convex optimization that we are aware of have utility bounds that degrade as $T \to \infty$** since the per-iteration privacy noise $\sigma^2$ blows up to $\infty$, drowning out the signal.
> > > >
> > > > >[Algorithm’s utility] “The final model parameters are chosen by drawing uniformly at random from all intermediate models during training. Won’t that often destroy model utility, if an early iterate is chosen?” This question was originally raised by Reviewer bxYf. In the rebuttal, the authors wrote “For example, we are not aware of any results that prove convergence in expectation for the last iterate of SGD (although almost sure convergence does hold).” However, the proposed algorithm in this paper is more like DP-SGD (or SGLD) rather than SGD. Establishing convergence guarantees for DP-SGD is easier than SGD because the Gaussian noise can potentially help the algorithm escape saddle points. I include one reference below...
> > > >
> > > > **The reference [Raginsky et al 2017] does not use noisy SGD and does not bound the stationarity gap**. Instead, they use stochastic gradient Langevin dynamics (SGLD) and bound excess risk. Note that SGLD is not equivalent to Gaussian noisy SGD. This can be seen by recognizing that as $T \to \infty$, SGLD is equivalent to the exponential mechanism of McSherry & Talwar (2007), which is $\epsilon$-DP; by contrast, as $T \to \infty$ in Gaussian noisy SGD, the resulting algorithm is not $\epsilon$-DP (since even the first iterate is not $\epsilon’$-DP for any $\epsilon’ < \infty$).
> > > >
> > > > The authors are well aware of the literature on escaping saddle points. **The literature on escaping saddle points is not related to stationary gap in private first-order optimization**. Our work is not concerned with escaping saddle points (as we are not talking about related concepts such as strict saddle, etc); we aim at only making the gradient norm small while maintaining privacy. This is the first step in understanding (private) algorithms in the nonconvex domain.
> > > > **We want to reiterate again that “we are not aware of any results that prove convergence [of stationarity gap] in expectation for the last iterate of SGD [or noisy SGD]”.** If you know of any such results, please let us know and we would be happy to study that and see if similar analysis can be performed in our context under privacy constraints.
> > > >
> > > > > there is a line of work in DP literature studying privacy amplification by iteration.
> > > >
> > > > Works on privacy amplification by iteration only consider **convex losses**. Their techniques and analysis rely heavily on convexity. Thus, they do not seem to be applicable in the non-convex setting that we consider in this work.  Finally, while it is not directly related to our paper, we are interested in learning such extensions to nonconvex loss functions if they exist. Please let us know if you are aware of such extensions to nonconvex losses.
> > > >
> > > > >[Fairness measures]...I think it is worth making it clear what kind of group fairness metrics cannot be covered by the analysis of this paper. For example, as the authors acknowledged in the rebuttal, false discovery rate and calibration error cannot be covered. In fact, some existing frameworks can incorporate these metrics into their framework…
> > > >
> > > > Thank you for letting us know. We will reference the works you mention and state that they can cover false discovery rate and calibration error while our work does not cover these metrics.

---

> ### Author Response · Authors · 2022-12-06
> **Have we addressed your concerns/questions?**
>
> We know it is a busy time of the year for everyone. We were wondering if we've addressed your concerns with our responses. Please let us know if any questions remain.
>
> Thank you,
>
> Authors

---

### Official Review · Reviewer_yWF7 · 2022-10-18

**Confidence:** 3
**Correctness:** 3
**Technical Novelty And Significance:** 2
**Empirical Novelty And Significance:** 3
**Recommendation:** 6

**Clarity, Quality, Novelty And Reproducibility:**

I think the previous section answered this. I appreciate that the authors have submitted their experiment code. I did not attempt to check the derivations in the appendix.

**Strength And Weaknesses:**

Strengths: The DP-FERMI formulation seems like a neat idea for getting to a convergence analysis. The paper is generally easy to follow (though there is a sudden drop-off in clarity when explaining technical bits, see below). The empirical performance looks like a nontrivial improvement over existing baselines.

Weaknesses: The application of FERMI is not obviously a large improvement over its introduction in (Lowy+ 2021): we want to optimize a weird fairness-constrained loss, so we instead optimize an upper bound on it, which admits a stochastic convergence analysis, and also handles non-binary classification. The added contribution here is, in the paper's phrasing, "a careful analysis of how the Gaussian noises [necessary for DP-SGD] propagate through the optimization trajectories". I don't have much feel for what constitutes an "interesting" convergence analysis, but the conceptual novelty here is unclear, and the introduction is a bit slippery about what is novel and what is borrowed from the FERMI paper.

The paper also struggles to explain its technical contributions in terms between a very high level summary and a long, opaque theorem statement. I suggest changing the focus of the paper to 1) reduce, relegate to the appendix, or eliminate the discussion of demographic parity (an extremely coarse notion of fairness that, IMO, the fairness literature needs to move past, and has only been discussed this long because it's very simple), which takes up over a page of the main body without meaningfully adding to the story told by the equalized odds results alone, 2) extending the discussion of how Theorem 3.1 works and what it accomplishes (the current statement is a blizzard of notation with little explanation -- I still don't know what W is doing), along with Theorem 3.2, and 3) extending the equalized odds results to more datasets (why are Parkinsons and Retired Adult results only reported for demographic parity? it seems like equalized odds should also apply here, and an empirical story built on 2 datasets seems thin). I think 2) would help provide a clearer explanation of the paper's improvement over (Lowy+ 2021) and 3) would make a stronger empirical case separate from the convergence analysis.

Other questions/comments:
1) I'd appreciate a table in the appendix attempting to concisely explain all of the relevant variables -- by my count, Theorem B.1 has well over a dozen.
2) Why is Tran 2021b a single point where the other methods have curves? More generally, perhaps I missed the explanation of this in the text, but what is varied to generate the curves?
3) As far as I can tell, the paper does not discuss the tightness of the upper bound offered by ERMI, nor does it explicitly write out the expression for equalized odds. This makes it hard to contextualize the convergence guarantee in terms of the underlying loss we actually want to optimize.
4) Figure 4 "priavte"

**Summary Of The Paper:**

The paper studies how to learn models that satisfy various fairness desiderata (e.g., demographic parity and equalized odds) while satisfying differential privacy. Like prior work, it writes down a loss function encoding some weighted combination of accuracy and fairness constraints and aims to develop a private method for optimizing it. The proposed "DP-FERMI" solution does so using DP-SGD. The claimed novelty lies in its convergence analysis, which appears to be the first of its kind for DP-SGD applied to a fairness-constrained loss. The key technical step relies on the "exponential Renyi mutual information" loss developed by (Lowy+ 2021) and used in that paper to derive a similar convergence result for stochastic/mini-batch (non-DP) SGD. While other papers have studied similar DP-SGD-based approaches to optimizing a fairness-constrained loss, those works did not attempt to theoretically analyze convergence. This paper concludes with some experiments demonstrating that DP-FERMI obtains better empirical tradeoffs between fairness and accuracy loss at the same privacy levels as existing baselines.

**Summary Of The Review:**

Overall, I think the current submission has some promising ideas and results that are hindered by an unclear explanation of technical or conceptual novelty and a sparse empirical evaluation on meaningful notions of fairness. As a result, my current recommendation is weak reject. However, I think a modified version of this paper, perhaps implementing the suggestions above, could be acceptable at ICLR or a similar venue, especially since the DP fairness literature has (IMO) still not featured much compelling work.

EDIT: After reading the author responses and discussing with some of the other reviewers, I've increased my score to a weak accept. Other reviewers have expressed stronger misgivings about 1) the assumptions required to set the parameters to satisfy the utility guarantee, 2) the difference between the "random iterate" version analyzed and the "final iterate" version implemented, and 3) the division between sensitive and non-sensitive attributes. IMO, these aren't serious problems, but after discussing with the other reviewers, I'm not planning to champion this paper further. I suggest that the authors attempt to address these issues and re-submit a modified version of this paper.

---

> ### Author Response · Authors · 2022-11-19
> **Response to Reviewer yWF7, Part 1.**
>
>
> Thank you for your thoughtful review and constructive feedback! We respond to each of your points below. Due to character limits, we split our response into two parts.
>
> >…not obviously a large improvement over [Lowy+ 2021]...
>
> The algorithm in Lowy et al. (2021) is not private. In settings where privacy is important or even mandated by law (e.g. in the European Union, due to GDPR), our algorithm constitutes a major improvement since the algorithm of  Lowy et al. (2021) would not even be able to be (legally) deployed. Notice that making algorithms private (while maintaining their performance to the extent possible) is not an easy task. There are many papers devoted to such tasks in various problems.
>
> > The added contribution here [compared to Lowy+2021] is, in the paper's phrasing, "a careful analysis of how the Gaussian noises [necessary for DP-SGD] propagate through the optimization trajectories". I don't have much feel for what constitutes an "interesting" convergence analysis, but the conceptual novelty here is unclear...
>
> Our main novel technical contribution is Theorem 3.2: the first privacy and convergence guarantee for differentially private nonconvex min-max optimization. Privately solving min-max problems is challenging and interesting, as evidenced by several recent papers devoted entirely to DP min-max optimization problems $\min_{\theta} \max_{W} F(\theta, W)$ in the easier/more restrictive cases of (strongly) convex or Polyak-Łojasiewicz $F(\cdot, W)$ (Boob & Guzmán, 2021; Yang et al., 2022; Zhang et al., 2022). As further evidence of the challenges associated with DP min-max optimization, note that while the sample complexity of DP convex optimization was settled back in 2014 (Bassily et al., 2014), DP convex-concave min-max problems remained a mystery until 2021 (Boob & Guzmán, 2021; Zhang et al., 2022).  **Our work is the first to prove that a non-trivial bound on the stationarity gap can be obtained for the more challenging DP min-max problem in which $F(\cdot, W)$ is an arbitrary *non-convex* smooth function.** We believe this contribution will be of independent interest to the DP optimization community.
>
> Although our convergence proof boils down to "a careful analysis of how the Gaussian noises [necessary for DP-SGDA] propagate through the optimization trajectories” (at a high level), properly executing this analysis and appropriately choosing algorithmic parameters to ensure convergence is non-trivial, as can be seen from the proof of Theorem 3.2.
> Additionally, our novel analysis even leads to sharper bounds for *non-private* SGDA compared with Lin et al. (2020), since our bound in Theorem B.1 has correct units and gives the precise dependence on smoothness with respect to $W$ and $\theta$ separately.
>
> > The paper also struggles to explain its technical contributions in terms between a very high level summary and a long, opaque theorem statement….I suggest extending the discussion of how Theorem 3.1 works and what it accomplishes (the current statement is a blizzard of notation with little explanation -- I still don't know what W is doing), along with Theorem 3.2…would help provide a clearer explanation of the paper's improvement over (Lowy+ 2021)
>
> Thank you for pointing this out! We shorten and simplify the statement of Theorem 3.1 in the revision. We did not want to spend too much space discussing Theorem 3.1 since it is due to Lowy et al. (2021). The key takeaway from Theorem 3.1 is that it allows us to use stochastic optimization to solve Equation (FERMI obj.) by re-formulating  (FERMI obj.) as a finite-sum min-max problem. The min-max formulation provides us with statistically unbiased estimators of (FERMI obj.). We mention this at the bottom of page 4.
>
> The main implication of Theorem 3.2 is that Algorithms 1 and 2 make the norm of the gradient of $FERMI(\theta)$ (or any other smooth non-convex objective function) small, where small is precisely quantified in terms of the number of samples and model parameters and the desired privacy level. Note that making the gradient small is generally the best one can hope to do in polynomial time for nonconvex optimization (Murty, 1985). Fortunately, the approximate stationary points that noisy gradient descent-based algorithms such as Algorithms 1 and 2 converge to are often approximate local minimizers as well (see e.g. Jin et al., 2017). We’ve added this discussion in the revision.
> Please let us know if there are any other specific points that you’d like us to discuss regarding Theorems 3.1 and 3.2.

---

> > ### Author Response · Authors · 2022-11-19
> > **Response to Reviewer yWF7, Part 2.**
> >
> > >I suggest changing the focus of the paper to 1) reduce, relegate to the appendix, or eliminate the discussion of demographic parity (an extremely coarse notion of fairness that, IMO, the fairness literature needs to move past, and has only been discussed this long because it's very simple), which takes up over a page of the main body without meaningfully adding to the story told by the equalized odds results alone
> >
> > As you correctly pointed out, demographic parity is appealingly simple. Thus, using demographic parity to explain our approach simplifies the presentation. The extension to equalized odds is straightforward once the reader understands how our algorithm works in the demographic parity case. It would actually take up more space to present our approach if we used equalized odds instead of demographic parity.
> >
> > >I suggest…extending the equalized odds results to more datasets (why are Parkinsons and Retired Adult results only reported for demographic parity? it seems like equalized odds should also apply here, and an empirical story built on 2 datasets seems thin)...would make a stronger empirical case separate from the convergence analysis.
> >
> > We agree with you and we have given some initial results of the equalized odds regularizer on the Retired Adult dataset and compared our results with Tran et al. tuned for the same dataset. We are currently working on tuning the non-private and/or the non-fair baseline for this framework to compare against the existing results. We are also tuning the Jagielski et. al model for this dataset too. Please see Appendix E in our rebuttal revision for these results.
> >
> > >Why is Tran 2021b a single point where the other methods have curves? More generally, perhaps I missed the explanation of this in the text, but what is varied to generate the curves?
> >
> > Tran. et. al. 2021b has a single point as **they solve the fairness constrained optimization in a primal-dual manner, which does not allow them to tune the tradeoff parameter** between the task dependent loss function and the loss based on the fairness violation. By contrast, our loss consists of two terms containing the task dependent loss function along with a **fairness regularizer with a coefficient ($\lambda$ in Theorem 3.1), providing a tradeoff between the fairness violation and misclassification error**. So, when we increase the tradeoff parameter, our model becomes more fair at the cost of accuracy. Hence, to explicitly visualize this tradeoff, we take different values of $\lambda$ and calculate the misclassification error and the fairness violation. We plot these accuracy-fairness points obtained for different values of $\lambda$ to obtain our curves.
> >
> > >As far as I can tell, the paper does not discuss the tightness of the upper bound offered by ERMI…
> >
> > While tightness of the upper bound offered by ERMI is theoretically interesting, we consider this to be outside the scope of our paper. The important point is that ERMI upper bounds other fairness violation notions: this implies that an algorithm which makes ERMI small will also make other fairness violation notions small. This is also supported empirically, where fairness violation is not measured in terms of ERMI and yet DP-FERMI outperforms the SOTA baselines.
> >
> > >[the paper does not] explicitly write out the expression for equalized odds [ERMI]. This makes it hard to contextualize the convergence guarantee in terms of the underlying loss we actually want to optimize.
> >
> > We’ve added this as Appendix B in the rebuttal revision.
> >
> > >Figure 4 "priavte"
> >
> > Fixed. We really appreciate your detailed comments in helping us improve the manuscript.

---

### Official Review · Reviewer_bxYf · 2022-10-18

**Confidence:** 3
**Correctness:** 3
**Technical Novelty And Significance:** 4
**Empirical Novelty And Significance:** 2
**Recommendation:** 8

**Clarity, Quality, Novelty And Reproducibility:**

The paper is mostly clear, novel, and reproducible. What does the width of the curves in the experimental figures represent? If it is a confidence interval, I am surprised that it is not wider due to the randomness of the final model choice. There is no baseline for the DCNN experiment. I understand that the Tran (2021b) doesn't work, but I would still like to see non-private or non-fair baseline performance.

**Strength And Weaknesses:**

Strength: It is maybe of theoretical interest that SGD-style algorithm is proposed for fair, private learning, with provable convergence even on non-convex loss.
Weaknesses:
1) The input of each training example is partitioned into sensitive features s and nonsensitive features x. The algorithm is private and fair with respect to s, while the trained classifier only uses x as input. It is not clear to me why one would want to partition features in this way. If I'm training a language model on text input I want my credit card number to be private, but I'm not worried about fairness with respect to credit card numbers. Conversely, I want my face recognition system to be fair with respect to skin tone, but I don't necessarily need that to be kept private. Finally, in principle it seems like it should be possible to use sensitive features as model input while retaining privacy and/or fairness.
2) The final model parameters are chosen by drawing uniformly at random from all intermediate models during training. Won't that often destroy model utility, if an early iterate is chosen?
3) The experimental results are not impressive. Other than Figure 3a, the results are not significantly stronger than the Tran et al. (2021b) baseline, although they do allow an interpolation between fairness and model utility. I would like to see non-private and/or non-fair baselines. How much model utility are we losing due to weakness #2 above?

**Summary Of The Paper:**

An SGD-style algorithm is proposed for fair, private learning, with provable convergence even on non-convex loss.

**Summary Of The Review:**

The described algorithm has the practical weaknesses I mentioned above. Unless I am mistaken, it seems unlikely that this method would be useful in practice. But as far as I know the technical contribution is novel and could be a useful stepping stone toward a more practical algorithm with the strengths of the proposed approach.

---

> ### Author Response · Authors · 2022-11-19
> **Response to Reviewer bxYf**
>
> Thank you for your thoughtful review and constructive feedback! We respond to each of your points below:
>
> >The algorithm is private and fair with respect to s, while the trained classifier only uses x as input. It is not clear to me why one would want to partition features in this way…in principle it seems like it should be possible to use sensitive features as model input while retaining privacy and/or fairness.
>
> This is a great point! We followed the standard setup in the DP fair learning literature going back to Jagielski et al. (2019) which only seeks to privatize the sensitive attributes. The motivation for this setup is that we want to train a fair model, but might not be able to use the sensitive attributes (which are needed to train a fair model) due to privacy concerns/laws (e.g. the EU’s GDPR). The solution proposed by Jagielski et al. (2019) is to ensure the differential privacy of the sensitive data, so that the company can safely use these sensitive attributes to train a fair model, while satisfying legal and ethical constraints on user privacy.
>
> That being said, you make a very valid point: there are certainly cases where we might want/need to ensure the privacy of features other than just the sensitive attributes. Fortunately, **our algorithm and analysis easily extends to the case where all the features are private**. To ensure differential privacy for all the features, the only change is that the noise variance $\sigma^2$ in Algorithm 2 would need to increase because the *sensitivity* estimates for the gradient updates would increase. Our stochastic Algorithm 2 still converges at the same asymptotic rate given in Corollary 3.1 when $d_{\theta} \gg kl$, as is typical in practical large-scale fairness problems; the rate only gets slower by constant factors and factors depending on $k$ and $l$. We are currently working on typing this up in the revision and including this extension in the appendix.
>
> >The final model parameters are chosen by drawing uniformly at random from all intermediate models during training. Won't that often destroy model utility, if an early iterate is chosen?
>
> **Choosing a random iterate is standard for theoretical analysis in nonconvex optimization**: see e.g., (Nesterov, 2004; Ghadimi & Lan, 2013; Carmon et al, 2016; Fang et al., 2018/2019; Carmon et al., 2020). Getting convergence guarantees for the last iterate is much harder. For example, we are not aware of any results that prove convergence in expectation for the last iterate of SGD (although almost sure convergence does hold). However, we always use the last iterate in practice (as it is common in most training procedures of nonconvex models). We agree that this is an interesting-to-study gap between theory and practice.
>
> **In our numerical experiments, we simply return the last iterate of our algorithm**. In the revision, we added a sentence clarifying that we recommend using the last iterate in practice.
>
> >Experimental results…other than Figure 3a, the results are not significantly stronger than the Tran et al. (2021b) baseline, although they do allow an interpolation between fairness and model utility. I would like to see non-private and/or non-fair baselines.
>
> This is not correct if you look at the experiments more closely. In fact, our algorithm offers significant improvements in fairness/accuracy tradeoffs over baselines in most experiments. To quantify the improvement of our results over Tran et al. (2021b) baseline, we calculated the performance gain with respect to fairness violation (for fixed accuracy level) that our model yields over all the datasets. **We obtained a performance gain of demographic parity that was 79.648% better than  Tran et al. (2021b) on average, and 65.89% better on median. The average performance gain of equalized odds was 96.65% while median percentage gain was 90.02%.**
>
> We have added the non-private and/or non-fair baselines in all of our plots in the revision.

---

> ### Comment · Reviewer_bxYf · 2022-11-21
> **Raised score**
>
> The authors have addressed my concerns and those of the other reviewers to my satisfaction. I have raised my recommendation accordingly.

---

> > ### Author Response · Authors · 2022-11-21
> > **Thank you**
> >
> > Thank you very much for your feedback. We are glad that you are satisfied with our responses and revision.

---

### Official Review · Reviewer_oW3z · 2022-10-26

**Confidence:** 4
**Correctness:** 3
**Technical Novelty And Significance:** 3
**Empirical Novelty And Significance:** 2
**Recommendation:** 5

**Clarity, Quality, Novelty And Reproducibility:**

The paper is well written. The novel contributions are specified, but, as stated above, the contributions need to be toned down substantially.


**Strength And Weaknesses:**

*Settings and contributions*:

The paper tackles an interesting and significant problem. I believe this one posed is a highly relevant question and am happy the authors studied this problem.

I have also, however, found the claimed contributions to be a bit overstated: It is nice that the proposed method allows to handle non-binary sensitive attributes and that multiple fairness definitions, but this is also what is currently assumed in the recent literature, including in some of the work against this paper compare.

I also believe that the claimed guarantees are overstated. From the abstract all the way to section 3 the paper claims that this proposal allows convergence for any fairness level even when mini-batches of data are used in each iteration of training.
This is, however, misleading. In section 3 and Theorem 3.1, in fact, after learning the assumptions adopted, I am left with the impression that claimed results are much less general than what is implied in the abstract and introduction: The restrictions taken on the loss function (strongly concave in the parameter space, global L-Lipschitz property in both the parameter of the primal and the dual space and convexity of the {\cal W} set.).

The claimed mechanism properties thus, need to be tuned down substantially in the eyes of this reviewer.
I encourage the authors to be very clear from the abstract on which assumptions the claimed properties are satisfied.

Notice that, related to this overstatement, the last sentence before section 4 "no existing work with convergence guarantees are able to handle non-binary classification" is misleading. I don't think it does good to our community.

*Experiments*:

I really like the way the authors have displayed the experiments. They show a good Pareto boundary and can clearly see the superiority of the proposed method w.r.t. Tran et al.

However, I also need to point out that there is no need to "adapt" Tran et al algorithms to non-binary cases, as their paper clearly reports experiments on non-binary cases too. I also found the statement about the instability of these methods to be possibly unjustified. Has a hyperparameter tuning over these methods been performed? Or did the authors use the hyperparameters adopted by Tran et al in their paper? If the latter is true, note that the datasets and architectures adopted for the experiments in this paper differ from what was used by Tran et al.


**Summary Of The Paper:**

The paper proposes a differentially private and fair learning algorithm, satisfying two common group fairness notions: demographic parity and equalized odds. The proposed algorithm relies on augmenting the standard empirical risk loss component with an exponential Renyi mutual information term between the protected group feature and the label. The latter term requires fine-tuning to fit the specific fairness notion adopted.
The claimed benefit of the proposed procedure, which relies on approximating the resolution of a min-max problem through a classic stochastic gradient method, is its convergence property even when this optimization procedure operates over mini-batches.


**Summary Of The Review:**

The paper develops a technique for an important much-needed problem: guaranteeing both privacy and fairness while providing convergence guarantees.
Some of the contributions of the work are however overstated and some clarity over the experimental setting adopted is needed.

---

> ### Author Response · Authors · 2022-11-19
> **Response to Reviewer oW3z**
>
> Thank you for your thoughtful review and constructive feedback! We respond to each of your points below:
>
> > I…found the claimed contributions to be a bit overstated: It is nice that the proposed method allows to handle non-binary sensitive attributes and that multiple fairness definitions, but this is also what is currently assumed in the recent literature…sentence “no existing work with convergence guarantees are able to handle non-binary classification" is misleading.
>
> We understand your concern about the possibility of overclaiming the contributions. We were also very careful about that. In particular, **to the best of our knowledge, the sentence “no existing [DP fair learning] works with convergence guarantees are able to handle non-binary classification" is accurate**. We did a thorough literature search and believe our claim is correct. Please let us know if you are aware of any previous works that provide such guarantees. Note that “convergence guarantee” here refers to a *provable* theoretical guarantee, not just convergence that is observed empirically in a particular experiment. We have clarified this in our revision (Figure 1 caption).
>
> Also, **we were careful about not claiming to be the only work that can handle non-binary sensitive targets/sensitive attributes and multiple fairness notions**. However, we claimed that we are the only work with a provably **convergent** algorithm that can handle non-binary targets: the only previous fairness algorithm with a (non-stochastic) convergence guarantee (Mozannar et al., 2020) could not handle non-binary target (see Figure 1).
>
> > …the paper claims that this proposal allows convergence for any fairness level even when mini-batches of data are used in each iteration of training. This is, however, misleading. In section 3 and Theorem 3.1, in fact, after learning the assumptions adopted, I am left with the impression that claimed results are much less general than what is implied in the abstract and introduction: The restrictions taken on the loss function (strongly concave in the parameter space, global L-Lipschitz property in both the parameter of the primal and the dual space and convexity of the {\cal W} set.).
>
> **We do not make any strong concavity assumption** to obtain our main theoretical result: privacy and convergence of DP-FERMI Algorithm 2 (Corollary 3.1). In fact, strong concavity (in W) of the fair ERM objective $\hat{F}(\theta, \cdot)$ automatically holds by (Lowy et al., 2021)---see the bottom of page 4.  This is the property of our regularizer, not the loss function.
>
> **Lipschitz continuity of the loss function is a standard assumption** in the differentially private optimization literature–see, e.g. (Chaudhuri et al, 2011; Bassily et al, 2014; Wang et al., 2017; Bassily et al., 2019; Feldman et al., 2020; Asi et al., 2021; Bassily et al., 2022) and the references therein. While we agree this could be a limiting assumption in general, we can always satisfy this assumption for differentiable loss functions with bounded data, as long as we limit the training domain to bounded parameter values.
>
> **We do not need to assume anything about the $\mathcal{W}$ set**. In fact, we get to choose this set in our algorithm. Thus, we can always choose it to be a sufficiently large (convex) ball, as in Corollary 3.1 so that we know it contains the optimal point.
>
> > Experiments…there is no need to "adapt" Tran et al algorithms to non-binary cases, as their paper clearly reports experiments on non-binary cases too. I also found the statement about the instability of these methods to be possibly unjustified. Has a hyperparameter tuning over these methods been performed? Or did the authors use the hyperparameters adopted by Tran et al in their paper? If the latter is true, note that the datasets and architectures adopted for the experiments in this paper differ from what was used by Tran et al.
>
> **Hyperparameter tuning over the baselines was performed** in order to account for the change in datasets and architectures, as correctly pointed out by the reviewer. We performed an exhaustive grid search on each of the baselines, obtaining the hyperparameters giving the best results for the baselines. We tuned the hyperparameters on data preprocessed in the same approach as Tran. el. al. The results for the baselines were also averaged over 15 runs on the best set of hyperparameters for each dataset at a certain fairness tradeoff value and privacy level. We have clarified this in our revision.
>
> **We would like to clarify that we did not adapt Tran. et. al to the large-scale case in terms of the methodology.** We used their original framework for that. In particular, we tuned many different hyperparameters of the models and tried various different vision-based architectures for the framework for the large-scale experiment for fairly evaluating the baseline. We have clarified this in our revision.

---

> > ### Author Response · Authors · 2022-12-06
> > **Have we addressed your concerns/questions?**
> >
> > We know it is a busy time of the year for everyone. We were wondering if we've addressed your concerns with our responses. Please let us know if any questions remain.
> >
> > Thank you,
> >
> > Authors

---

### Author Response · Authors · 2022-11-19
**Note to all reviewers**

Dear Reviewers,

Thank you for your thoughtful reviews and constructive, detailed feedback. We have incorporated your suggestions into the rebuttal revision, where textual changes are highlighted in violet. We have also revised and added some plots of our experimental results in the revision. We reply to each of your individual reviews below.

Sincerely,

Authors

---

### Decision · Program_Chairs · 2023-01-20

**Decision:**

Accept: poster

**Justification For Why Not Higher Score:**

Results may be a bit overstated, and are not strong enough to warrant a spotlight for the community.

**Justification For Why Not Lower Score:**

Results and problem interesting

**Metareview: Summary, Strengths And Weaknesses:**

This paper received significant discussion from the reviewers. Some concerns are as follows.
- Convergence guarantees: the parameters required for convergence may not be known to the data scientist
- Utility of theoretical versus empirical algorithm are different (one samples a random iterate, the other outputs the last)
- Utility guarantees find a stationary point rather than one of low loss
- Privacy is assumed to be for features iff they are ones we should be fair wrt

The first three concerns were felt to be common to several works in the literature, and reviewers and myself were divided on whether they are reasonable or not. In the end, I consider them to be "OK." I personally agree that the first may or may not be fully reasonable, but it is not necessarily the job of this paper to resolve it. I think the second and the third concerns are not an issue in this work or any of the others -- it is too high a standard for the theoretical algorithm to 100% match the empirical algorithm (if we did this, we would never get any interesting algorithms from either a theoretical or empirical side), and finding a stationary point is a design choice of what the optimization goal is. The authors may want to further discuss these restrictions and whether or not they are reasonable or not.

However, reviewers seemed to agree upon empirical improvement and the problem being interesting.

The authors may also want to address the concerns of Reviewer oW3z, who was the most critical and feels that the results are somewhat overstated. While their concerns were noted and justifiable, it was felt that these revisions could be addressed without a whole new round of submission. I asked the reviewer to elaborate in more detail of their concerns in their final review, and I trust that the authors will adequately address them to the extent that the reviewer clarifies.

**Note From Pc:**

if the above contains the word "oral" or "spotlight" please see: "oral" presentation means -> notable-top-5% and "spotlight" means -> notable-top-25%. As stated in our emails, we are disassociating presentation type from AC recommendations